# Prussian blue analog with separated active sites to catalyze water driven enhanced catalytic treatments

Liu-Chun Wang [1], Pei-Yu Chiou[1], Ya-Ping Hsu[1], Chin-Lai Lee[2], Chih-Hsuan Hung[1], Yi-Hsuan Wu[1], Wen-Jyun Wang[3], Gia-Ling Hsieh[1], Ying-Chi Chen[1], Li-Chan Chang [4], Wen-Pin Su [4,5], Divinah Manoharan[1], Min-Chiao Liao[2], Suresh Thangudu [6], Wei-Peng Li [3,7,8] ✉, Chia-Hao Su [6,9,10,11] ✉, Hong-Kang Tian [12,13,14] ✉ & Chen-Sheng Yeh [1,8] ✉

Chemodynamic therapy (CDT) uses the Fenton or Fenton-like reaction to yield toxic ·OH following $H_2O_2 \rightarrow$ ·OH for tumoral therapy. Unfortunately, $H_2O_2$ is often taken from the limited endogenous supply of $H_2O_2$ in cancer cells. A water oxidation CoFe Prussian blue (CFPB) nanoframes is presented to provide sustained, external energy-free self-supply of ·OH from $H_2O$ to process CDT and/or photothermal therapy (PTT). Unexpectedly, the as-prepared CFPB nanocubes with no near-infrared (NIR) absorption is transformed into CFPB nanoframes with NIR absorption due to the increased $Fe^{3+}$-N ≡ C-$Fe^{2+}$ composition through the proposed proton-induced metal replacement reactions. Surprisingly, both the CFPB nanocubes and nanoframes provide for the self-supply of $O_2$, $H_2O_2$, and ·OH from $H_2O$, with the nanoframe outperforming in the production of ·OH. Simulation analysis indicates separated active sites in catalyzation of water oxidation, oxygen reduction, and Fenton-like reactions from CFPB. The liposome-covered CFPB nanoframes prepared for controllable water-driven CDT for male tumoral mice treatments.

Chemodynamic therapy (CDT), a new modality of cancer treatment based on reactive oxygen species (ROS), uses Fenton or Fenton-like reactions to yield the toxic hydroxyl radical (·OH) following $H_2O_2 \rightarrow$ ·OH processing to selectively kill cancer cells while sparing healthy normal cells[1,2]. Usually, $H_2O_2$ is taken from the overproduction of $H_2O_2$

in cancer cells. Unfortunately, endogenous $H_2O_2$ is not sufficient to drive the reaction kinetics for sustainable ROS production for CDT due to the antioxidative mechanism in the cellular function[3]. A tumor microenvironment-responsive catalytic strategy is proposed to boost $H_2O_2$ concentrations. Because of the Warburg effect, glucose is more

[1]Department of Chemistry, National Cheng Kung University, Tainan 701, Taiwan. [2]Department of Diagnostic Radiology, Kaohsiung Chang Gung Memorial Hospital, Kaohsiung 833, Taiwan. [3]Department of Medicinal and Applied Chemistry, Kaohsiung Medical University, Kaohsiung 807, Taiwan. [4]Institute of Clinical Medicine, College of Medicine, National Cheng Kung University, Tainan 704, Taiwan. [5]Departments of Oncology and Internal Medicine, National Cheng Kung University Hospital, College of Medicine, National Cheng Kung University, Tainan 704, Taiwan. [6]Center for General Education, Chang Gung University, Taoyuan 333, Taiwan. [7]Drug Development and Value Creation Research Center, Kaohsiung Medical University, Kaohsiung 807, Taiwan. [8]Center of Applied Nanomedicine, National Cheng Kung University, Tainan 701, Taiwan. [9]Department of Biomedical Imaging and Radiological Sciences, National Yang Ming Chiao Tung University, Taipei 112, Taiwan. [10]Department of Radiation Oncology, Kaohsiung Chang Gung Memorial Hospital, Kaohsiung 833, Taiwan. [11]Institute for Radiological Research, Chang Gung University, Taoyuan 333, Taiwan. [12]Department of Chemical Engineering, National Cheng Kung University, Tainan 701, Taiwan. [13]Program on Smart and Sustainable Manufacturing, Academy of Innovative Semiconductor and Sustainable Manufacturing, National Cheng Kung University, Tainan 701, Taiwan. [14]Hierarchical Green-Energy Materials Research Center, National Cheng Kung University, Tainan 701, Taiwan. ✉e-mail: wpli@kmu.edu.tw; chiralsu@gmail.com; hktian@gs.ncku.edu.tw; csyeh@mail.ncku.edu.tw

abundant in cancerous cells than in normal cells[4]. Accordingly, glucose oxidase, a natural dehydrogenase, is incorporated with nanoparticle (NP) carriers to catalyze the intrinsic glucose in tumor cells into $H_2O_2$[5–7]. However, this elegant approach still relies on the presence of endogenous glucose. Furthermore, the incorporation of glucose oxidase into NP carriers increases complications of nanoformulation. Recently, metal peroxide nanoparticles (NPs) have demonstrated decomposition behavior that achieves self-supply of $H_2O_2$, thereby exerting an anticancer effect[8,9]. However, these metal peroxides suffer from short-lived $H_2O_2$ production due to the rapid exhaustion of raw material. In addition, when the therapeutics applied to malignant tumors account for the $O_2 \rightarrow H_2O_2$ process, chemical drugs, i.e. cis-platin and doxorubicin, have been found to activate superoxide dismutase (SOD)-mediated redox reaction to generate $H_2O_2$ from $O_2$ in cancer cells.[10,11] However, this process is restricted to the treatment of hypoxic tumors. In this context, $O_2$ supplementation within cells has derived $O_2$ from the catalysis of $H_2O_2$ from catalase or $MnO_2$[12–14]. Overall, the supply of $H_2O_2$ is restricted in the endogenous tumor microenvironment. This raises the need to discover a material that would allow for the sustainable self-supply of $H_2O_2$.

Living tissues contain significant amounts of water which is used in many biological processes. This water could potentially be used directly as a source for continuous production of $H_2O_2$ through certain reactions, benefiting further Fenton or Fenton-like reactions for disease therapeutics. Based on this concept, it would be useful to develop a material which can simply use this tissue-based water for the sustainable self-supply of $O_2$ and $H_2O_2$. This would, in turn, simplify NP formulation for application in cancer treatments, benefitting clinical translation from bench to bedside. Notably, water-splitting, widely used for energy storage and environmental applications[15–17], has been used in CDT[18,19]. The $O_2$, $H_2$, and $\cdot$OH products of water-splitting have been used for various therapeutic processes[20]. However, water-splitting still requires the application of external energy (e.g., light, ultrasound or electricity) for successful oxidation[20].

Prussian blue (PB) NP constructed using a $Fe^{3+}$-N≡C-$Fe^{2+}$ based framework reveals unique absorption at the near-infrared (NIR) region attributed to the inter-charge transfer between $Fe^{2+}$ to $Fe^{3+}$[21]. However, there is no NIR absorption feature found from the PB analogs (PBAs) due to the absence of $Fe^{3+}$-N≡C-$Fe^{2+}$ composition. The NIR region belongs to the biological window where the irradiation in this region shows deeper tissue penetration and lower phototoxicity than in the UV and visible regions[22]. Taking this optical feature in NIR absorption, PB can act as an ideal photothermal agent for photothermal therapy (PTT)[23]. Moreover, the redox-active $Fe^{2+}$/$Fe^{3+}$ couples in PB also play a critical role in endowing the ability of ROS-scavenging[24–26]. PBAs prepared by replacing Fe with other metal species showed distinct characteristics[27]. For example, the Mn in the MnFe PBAs serves as a critical electron donor site to catalyze the peroxymonosulfate decomposition for ROS generation to degrade pollutants[28]. MnCo PBAs partially incorporating Ru atom revealed single-atom catalytic ability for oxygen production from endocellular $H_2O_2$[29].

Herein, we report the first example of stable water oxidation NPs, allowing for the sustainable, external energy-free self-supply of $\cdot$OH for CDT and/or PTT. CoFe Prussian blue (CFPB) nanocubes treated with proton-induced metal replacement reactions were unexpectedly transformed into CFPB nanoframes with ex nihilo near-infrared (NIR) absorption due to increased $Fe^{3+}$-N≡C-$Fe^{2+}$ units in the CFPB framework. Remarkably, both the CFPB nanocubes and nanoframes exhibit a unique ability to catalyze the water-initiated tandem reaction ($H_2O \rightarrow O_2 \rightarrow H_2O_2 \rightarrow \cdot$OH) for sustainable $O_2$, $H_2O_2$, and $\cdot$OH generation. In particular, the CFPB nanoframes outperform the nanocubes in terms of $\cdot$OH production, making them a promising catalytic agent for tumors. Simulation analysis indicates that the catalysis of water oxidation, oxygen reduction, and Fenton-like reactions were achieved by active site-separated CFPB nanocatalyst. Unlike Prussian blue NP, which is a ROS-scavenger[24–26], the current CFPB nanocubes and nanoframes can generate $\cdot$OH. Consequently, the resulting $\cdot$OH is used for CDT (Fig. 1a) and the NIR characteristic absorption from the nanoframes can be used for PTT as well. PB and PBAs have not been previously reported as activating the oxidation of water to generate $O_2$, $H_2O_2$, and $\cdot$OH without an external energy supply[18–22,30–34].

## Results

### Structural transformation of CFPB from nanocubes to nanoframes with growing NIR absorption ex nihilo

CFPB nanocubes were prepared as the starting material with basic characteristics analyzed by TEM, SEM, electron diffraction, EDX, XRD, UV–Vis, and FTIR showing monodispersed cubic NPs with an edge length of 112 nm, well-defined crystalline structures, and no characteristic absorbance in the NIR region (Fig. 1b and Supplementary Fig. 1).

The as-prepared CFPB nanocubes were subjected to acidic corrosion through treatment with HCl (0.01 M) at 90 °C under heating in an oil bath. Morphological changes during this process were monitored at various reaction times, as seen in TEM and SEM images (Fig. 1b–g). In the early stage (0.5 h), most of the nanocubes remain solid, but a few cubes exhibit bright edges suggestive of dissolution. Nanocube appearance changes to assume a concave surface (3 h), indicating the dissolution of the middle part of the faces. A hollow morphology, with a small cube located in the core of a nanoframe, begins emerging (6 h). Most of the solid cubes turn into hollow interiors with some nanoframes containing uneroded nanocubes (red arrows in 1 f). The completed CFPB nanoframes were obtained after 24 h of reaction (Fig. 1g). Tilting the nanoframes 30° under TEM provides additional evidence of the frame-like structures (Supplementary Fig. 2). Following composition characterization in various etching periods, the results in Co/Fe ratios were seen as a function of structural change by atomic absorption (AA) measurements (Fig. 1h and Supplementary Table. 1). The Co/Fe ratio decreased progressively from 1.42 to 0.20 after 24 h reaction, indicating the loss of cobalt ions during etching. Element mapping analysis reveals the distribution of cobalt and iron ions within the nanoframes (Fig. 1i). Regarding the optical features, the absorbance in the NIR region from 600 to 900 nm surprisingly increases over the reaction time and the color of the colloidal solutions changes gradually from brown to blue, strongly suggesting the production of $Fe^{3+}$-N≡C-$Fe^{2+}$ composition during acid etching (Fig. 1j). The blue color typical of PB originates from the charge transfer of $Fe^{2+}$ to $Fe^{3+}$. Because of the appearance of NIR absorption, the CFPB nanoframes could serve as a photothermal therapeutic agent in biomedical applications. The CFPB nanoframes (14 ppm in Co) exhibit a rapid temperature increase to 42 °C, conducive to the killing of cancer cells within a 5-min exposure time (Fig. 1k and Supplementary Fig. 3). The photothermal conversion efficiency $\eta$ of CFPB nanoframes under irradiation (808 nm) is 13.6% (see Supplementary). On the other hand, no apparent heating behavior is seen in the CFPB nanocubes (Fig. 1k).

### Structural characterization of CFPB during acid etching

As determined through TEM elemental analysis, the Co/Fe ratio of the selected intact particle (6 h) in the reaction is 0.59 (Fig. 2a), which is close to the 0.61 obtained through AA (Fig. 1h). Regarding the compositional distribution of the 6 h product, the edge regions produce ratios that are clearly lower (e.g., 0.28 and 0.4) than that of the center part (1.36). As for the 24 h product, the Co/Fe ratios range between 0.23 and 0.27 and are distributed consistently in the edges (Fig. 2b). Once again, the 0.3 Co/Fe ratio in the selected intact particles through TEM analysis of the 24 h-product approximates the 0.2 AA measurement results. The CFPB nanoframes, as characterized by high-resolution TEM and electron diffraction, retain a crystalline structure (Supplementary Fig. 4).

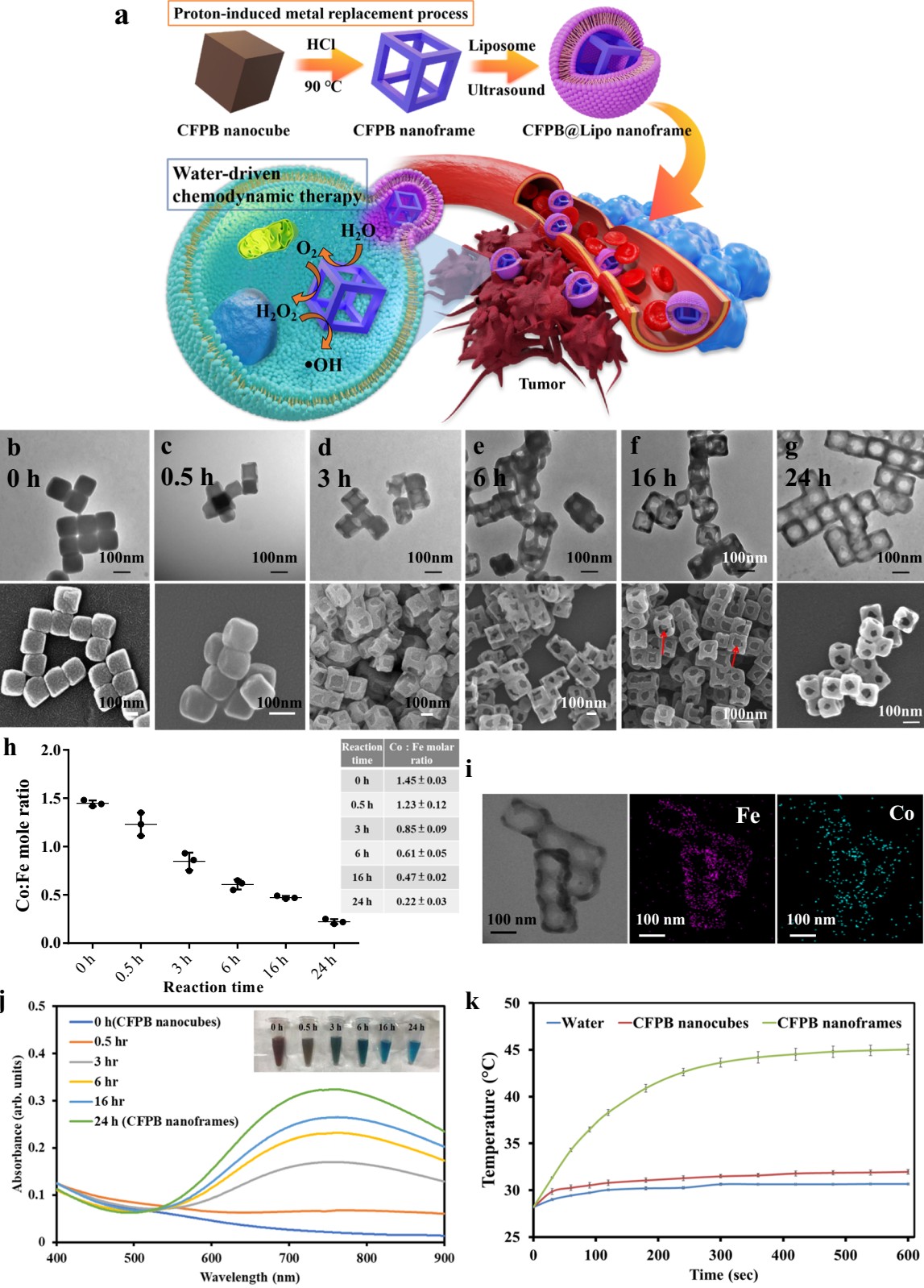

**Fig. 1 | Morphological evolution, optical feature, element mapping, and atomic analysis of CFPB during acid etching. a** Illustration of the structural changes of CFPB from nanocube to nanoframe and the controllable strategy to proceed water-driven CDT on cancer cells. **b–g** TEM and SEM images showing the structural transformation of CFPB. All scale bars are 100 nm. **h** Co/Fe ratios determined by AA measurements as a function of the etching duration. **i** Element mapping analysis of the nanoframes after 24 h of reaction. **j** UV–Vis spectrum of the nanostructures as a function of the etching duration. Inset: Photograph depicting the color of colloidal solutions following etching duration. **k** The heating performance of CFPB nanocubes and CFPB nanoframes under exposure to an 808-nm laser diode at 0.8 W/cm². All data were obtained in triplicate ($n = 3$, the error bars represented mean ± SD). Source data are provided as a Source Data file. **a–g**, **i**, **j** One representative data was shown from three independently repeated experiments.

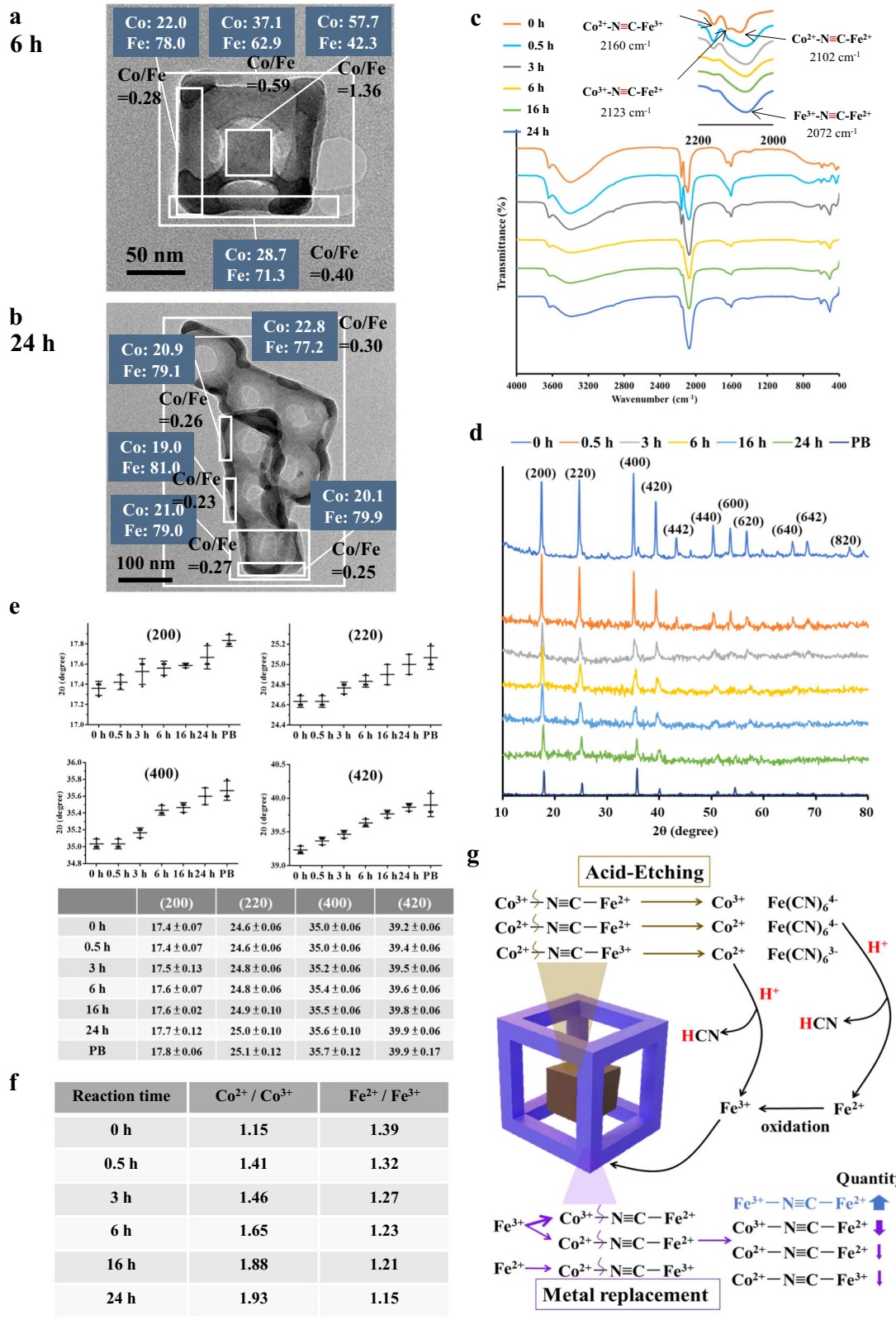

**Fig. 2 | Compositional analysis of CFPB during acid etching and the mechanism of the acid corrosion reaction. a** Area selected for the elemental analysis of the CFPB after 6 h of reaction. **b** Area selected for the elemental analysis of the CFPB after 24 h of corrosion. **c** FTIR spectra monitored as a function of the etching period with CFPB nanocubes as the starting material. **d**, XRD spectra monitored as a function of the etching period with CFPB nanocubes as the starting material. The diffraction peaks in black color indicate the patterns of pure PB nanocrystals. **e** The shift of the diffraction angle in 2θ showing change in the (200), (220), (400), and

(420) crystal planes monitored as a function of the etching duration with CFPB nanocubes as the starting material. **f** $Co^{2+}/Co^{3+}$ and $Fe^{2+}/Fe^{3+}$ ratios determined through XPS monitored as a function of the etching duration with CFPB nanocubes as the starting material. **g** Illustration of the acid corrosion following proton-induced metal replacement reaction mechanisms. All data were obtained in triplicate ($n = 3$, the error bars represented mean ± SD). Source data are provided as a Source Data file. **a**–**d** One representative data was shown from three independently repeated experiments.

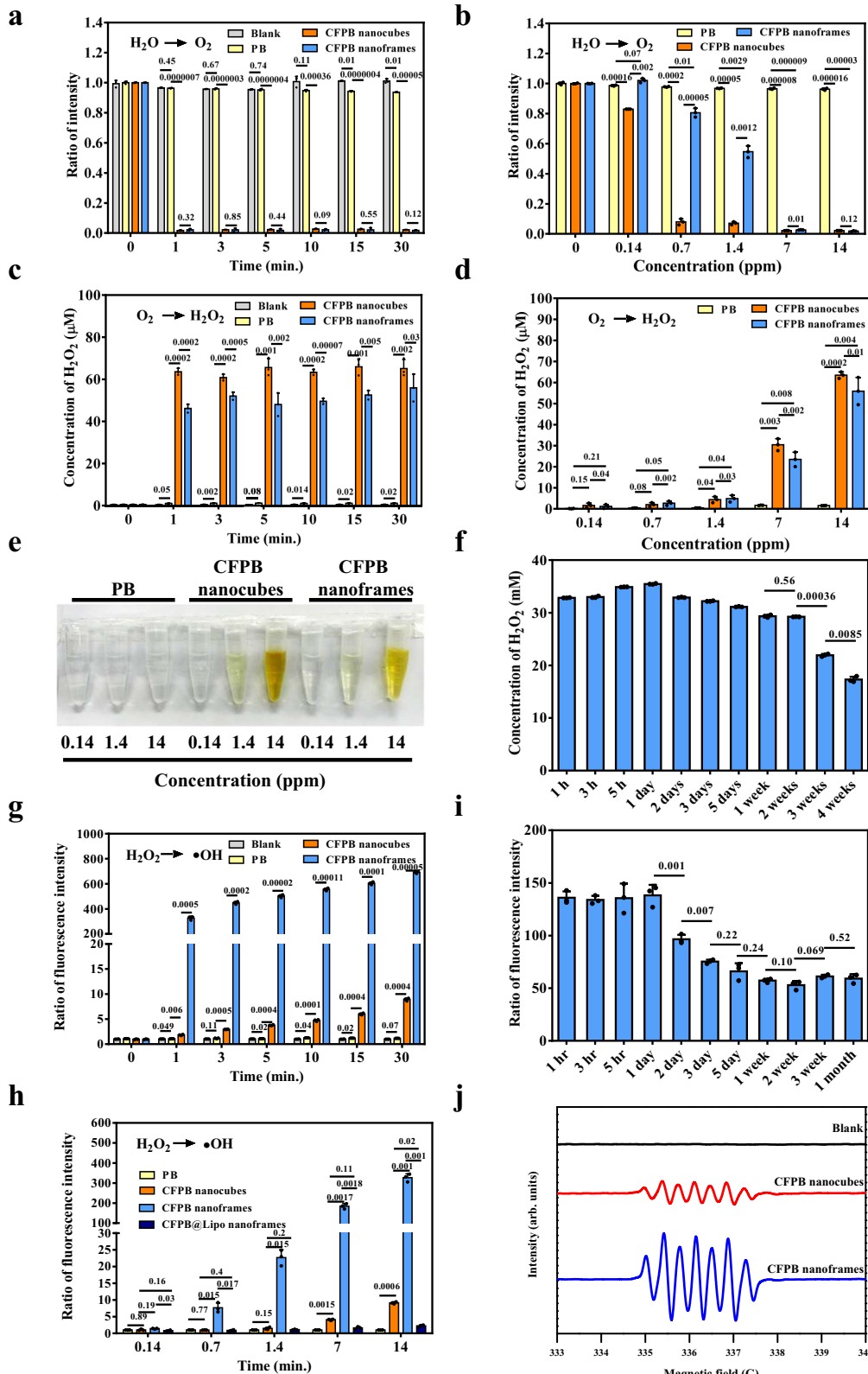

Further compositional characterization is conducted through FTIR, XRD, and XPS. FT-IR reveals the vibration signals of $Co^{2+}$-N≡C-$Fe^{3+}$, $Co^{3+}$-N≡C-$Fe^{2+}$, and $Co^{2+}$-N≡C-$Fe^{2+}$ respectively from the CFPB nanocubes at 2160, 2123, and 2102 cm$^{-1}$ (Fig. 2c). As the structural corrosion begins, these three vibrations gradually red-shift to a single peak at around 2072 cm$^{-1}$ after 24 h of reaction. Notably, pure PB NP has a representative vibration peak at 2072 cm$^{-1}$[35]. XRD patterns were

monitored as a function of reaction time (Fig. 2d, e). The diffraction angles in the (200), (220), (400), and (420) crystal planes were monitored as a function of the etching duration. All the diffraction peaks gradually shift to higher $2\theta$ angles, indicating a transition from a CFPB composition toward a PB composition. XPS analysis was conducted to evaluate variations in the oxidation states of cobalt and iron ions across reaction stages (Fig. 2f and Supplementary Figs. 5 and 6). The

**Fig. 3 | Self-supply generation of O₂, H₂O₂, and ·OH from H₂O from CFPB nanocubes and nanoframes. a, b** Oxygen generation from water oxidation activated by PB, CFPB nanocubes and nanoframes at a fixed as 14 ppm of Fe for PB and Co for CFPB for at various reaction times and at various cobalt concentrations over 30 min. Prior to O₂ measurement, O₂ was removed using a purge procedure with N₂. The dye of Ru(dpp)₃Cl₂, as an oxygen indicator, was employed to examine oxygen generation in 20-mL vials observing the decrease in fluorescence intensity at 613 nm in the presence of oxygen. **c, d** H₂O₂ generation activated by PB, CFPB nanocubes and nanoframes at a fixed as 14 ppm of Fe for PB and Co for CFPB for at various reaction times and at various cobalt concentrations over 30 min. A purge was conducted to remove O₂ from the test tube before the measurements. **e** Colorimetric analysis of H₂O₂ generation showing yellowish color from PB, CFPB nanocubes and nanoframes at various concentrations of Fe and cobalt. One representative data was shown from three independently repeated experiments.

**f** Continuous observation of H₂O₂ as a function of time for nanoframes. **g, h** The APF dye was used to determine the enhanced fluorescence at 520 nm in the presence of ·OH. Fluorescence intensity obtained from PB, CFPB nanocubes and nanoframes at a fixed as 14 ppm of Fe for PB and Co for CFPB for at various reaction times and at different concentrations of CFPB nanocubes, nanoframes, and CFPB@Lipo nanoframes over 1 min. The ratio of the fluorescence intensity was calculated from the APF dye without and with nanostructures. **i** Sustainable observation of ·OH generation from CFPB nanoframes at a fixed cobalt concentration of 1.4 ppm. **j** The generation of ·OH was evaluated by ESR spectrometer using 100 mM 5,5-Dimethyl−1-pyrroline N-oxide (DMPO) spin-trapping adduct with and without CFPB nanocubes and nanoframes (1.4 ppm of Co concentration). All data were obtained in triplicate ($n = 3$, The error bars represented mean ± SD). $p$-values were calculated by one-way ANOVA. Source data are provided as a Source Data file.

Fe²⁺/Fe³⁺ ratios gradually decrease from 1.39 to 1.15 over 24 h, suggesting an increase of Fe³⁺ in the structures. We also noticed an increase of Co²⁺/Co³⁺, indicating a drop in Co³⁺. From the results for morphological variations, optical changes, elemental characterization, and compositional identification, we suggest a proton-induced metal replacement reaction occurred in the etching process (Fig. 2g), which will be discussed later.

### Sequential reactions of H₂O → O₂ → H₂O₂ → ·OH for CFPB nanocubes and nanoframes in water

Unexpectedly, both CFPB nanocubes and nanoframes can generate O₂, H₂O₂, and ·OH from water. Specifically, the nanoframes outperform in the production of ·OH, making them a promising chemodynamic agent. Generally, CFPB can be triggered for electro- or photocatalysis for a water-splitting reaction to produce abundant oxygen[36]. The current CFPB nanocubes and nanoframes oxidize water to produce oxygen without an external energy supply. Prior to O₂ measurement, O₂ was removed from the test tube using a purge procedure with N₂. The O₂ fluorescent sensing agent [Ru(dpp)₃]Cl₂ was used to observe the decrease of fluorescence intensity at 613 nm in the presence of O₂. Rapid oxygen generation was observed within 1 min at a fixed cobalt concentration of 14 ppm (Fig. 3a). For comparison, PB showed no O₂ generation within 30 min. The concentration dependence indicates that the fluorescence dramatically drops at 0.7 ppm for the nanocubes while an apparent decrease appeared at 7 ppm for the nanoframes (Fig. 3b). The 0 min mark in Figs. 3a, b represents the fluorescence of the indicator only without NPs.

In general, metal can serve as an active site for the oxygen reduction reaction (ORR) under energy supply[37,38]. Surprisingly, both CFPB nanocubes and nanoframes perform this reaction readily, generating a high amount of H₂O₂ without external triggers. A colorimetric method is used to monitor the reaction between potassium iodide (KI) and H₂O₂ to yield yellowish I₃⁻ with absorbance at 350 nm followed by UV−Vis quantitation. Once again, O₂ was purged from the test tube prior to measuring. H₂O₂ production occurs rapidly within 1 min before reaching a plateau at a fixed cobalt concentration of 14 ppm (Fig. 3c). PB revealed no H₂O₂ generated within 30 min. The 0 min represents no NP addition. Concentration-dependent behavior was seen in both the nanocubes and nanoframes (Fig. 3d, e). We observed continuous H₂O₂ production by CFPB nanoframes for a month (Fig. 3f).

Co²⁺ is a well-known active catalyst for Fenton-like reactions to generate ·OH from H₂O₂[39]. Aminophenyl fluorescein (APF), a ·OH indicator, was used to monitor fluorescence increase from the PB, CFPB nanocubes and nanoframes as a function of reaction time and concentration (Fig. 3g,h). Once again, PB showed no ·OH production. The 0 min represents no NP addition. The concentration-dependent profile (Fig. 3h) was obtained after 1 min of reaction for the generation of ·OH. Notably, the nanoframes significantly outperform the nanocubes in ·OH formation, likely due to the larger surface area of the frame-like structure as well as the higher Co²⁺ content of the

nanoframes. Long-term (1 month) observation of ·OH generation from the nanoframes indicates a sustainable chemodynamic reaction (Fig. 3i). In fact, the nanoframes were found to retain their structure after 8 months of storage (Supplementary Fig. 7). Electron spin resonance (ESR) analysis further reveals ·OH generation from both the CFPB nanocubes and nanoframes where nanoframes show greater ESR signal amplitude (Fig. 3j).[40] The characteristic signals with an amplitude 1:2:2:1 quartet corresponding to DMPO-OH and the peaks for the dimer of DMPO composed by a characteristic 1:1:1 triplet. The excess ·OH can further oxidize DMPO-OH to form the paramagnetic dimers showing a triplet signal.

We present CFPB nanoframes that act as a promising water-driven chemodynamic nanodrug. However, increased oxidative stress makes blood circulation unfavorable to a sustained chemodynamic reaction. Thus, a lipid membrane was introduced to coat the CFPB nanoframes. Prepared liposomes were subjected to a sonication-cooling process to form CFPB@liposome (CFPB@Lipo). The zeta potential, hydrodynamic diameter, FTIR, TEM image and TGA analysis were also determined to indicate the successful modification of liposomes onto the CFPB nanoframes (Supplementary Fig. 8). The zeta potential of CFPB nanoframes is ~−32 mV. The CFPB@Lipo nanoframe surface becomes more positive (~8 mV) after Lipo modification, indicating the successful modification of Lipo onto the CFPB nanoframes. The hydrodynamic diameter apparently increases after Lipo modification. The vibration signals of FTIR spectra correspond to the bonds of C = O, C = C, C-N, P = O, P-O and C-O respectively observed at 1726, 1634, 1230, 1170, 1090 and 980 cm⁻¹ in the spectra of liposomes and CFPB@Lipo nanoframes given the liposome presence on the CFPB nanoframes surface. TEM imaging reveals CFPB@Lipo nanoframes. TGA analysis showing the remaining weights of 37.8% and 25.5% respectively for CFPB nanoframes and CFPB@Lipo nanoframes. From the weight loss, the quantification of Lipo was determined to be 12.3%. The phospholipid bilayer membrane with a hydrophobic component could act as a water-impenetrable barrier to efficiently switch off the water-driven catalytic reactions in the course of blood circulation. The high affinity of the lipid membrane to the cell membrane facilitates CFPB diffusion into the cells. The production of ·OH is significantly suppressed, suggesting that the lipid membrane coating can effectively prevent the continuous chemodynamic reaction during blood circulation prior to entry into cancer cells (Fig. 3h). The CFPB@Lipo nanoframes exhibit excellent stability, retaining intact structure after 7 days of incubation in various media (Supplementary Fig. 9).

### Simulation analysis of water-driven chemodynamic reactions

To understand the mechanism by which the CFPB catalyst promotes water oxidation/oxygen reduction, we investigated the electronic properties of CFPB via first-principles Density Functional Theory (DFT) calculations. Based on the essential feature of an active site-isolated catalyst, we propose that various crystal environments with specific Co-N ≡ C-Fe compositions at different regions of a CFPB catalyst may

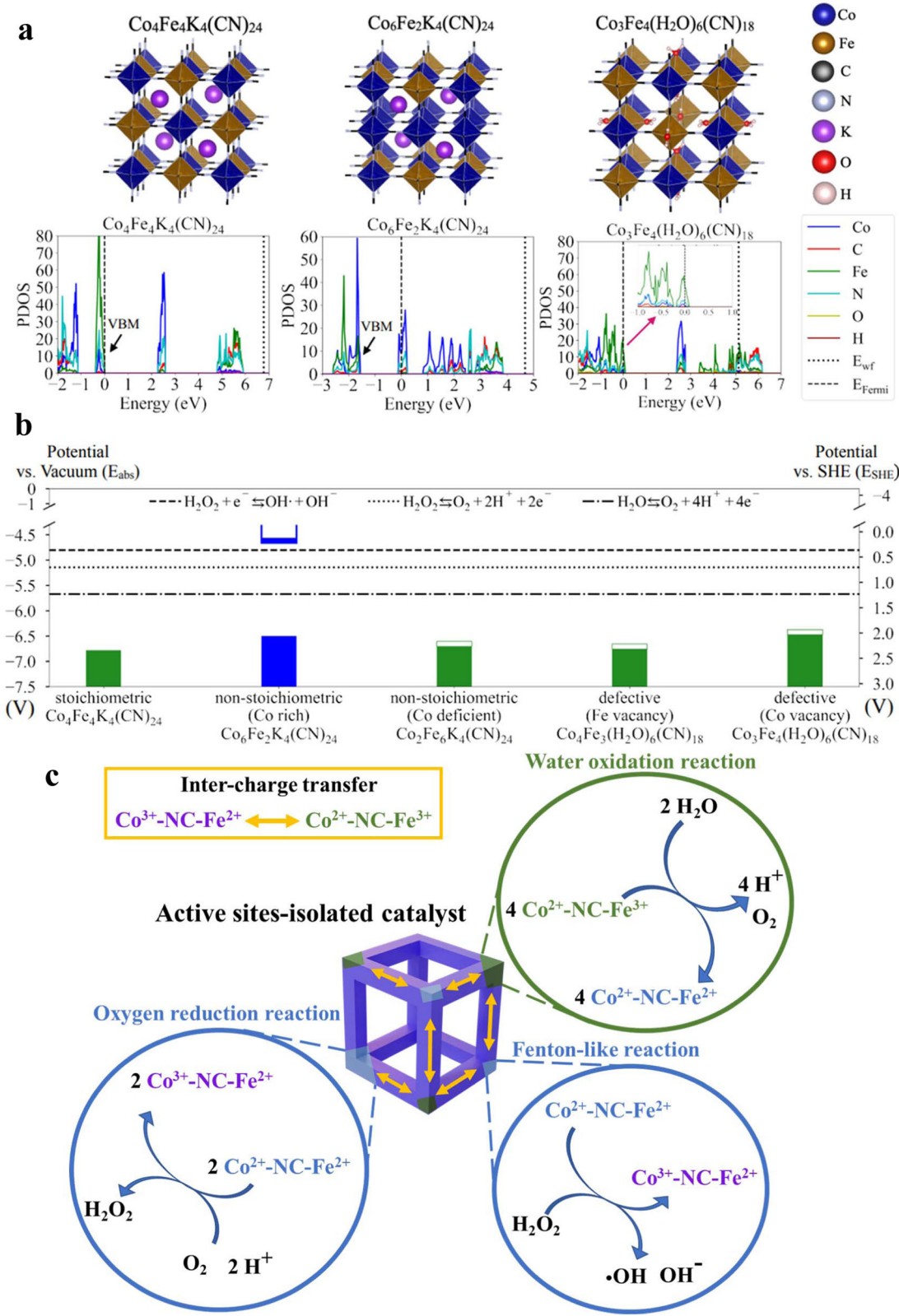

result in anisotropic electronic properties, allowing for tandem reactions[41,42]. Fig. 4a shows the atomic structures of different CFPBs with their calculated Projected Density of States (PDOS). The detailed computational methods and structure constructions can be found in the Method and Supplementary Information sections. The calculated band gap of 2.2 eV and the lattice parameter of 9.82 Å for stoichiometric CFPB are consistent with previous results[35,36,43]. PDOS results

show that the Valence Band Maximum (VBM) is mainly contributed by Fe, and the Conduction Band Minimum (CBM) is primarily composed of Co, indicating the existence of $Fe^{2+}$ and $Co^{3+}$ in stoichiometric CFPB.

The non-stoichiometric (Co-rich) CFPB shows a lifted Fermi level in the conduction bands, contributed by Co because of the extra electrons introduced by Co replacements. On the other hand, the Co-defective CFPB exhibits a downward Fermi level shift of 0.1 eV because

**Fig. 4 | First-principles calculations of CPFB bulk and surface structures and the resulting band alignments. a** Atomic bulk structures of stoichiometric $(Co_4Fe_4K_4(CN)_{24})$, non-stoichiometric Co rich $(Co_6Fe_2K_4(CN)_{24})$, and defective-Co vacancy $(Co_3Fe_4(H_2O)_6(CN)_{18})$ CPFB, and the corresponding calculated PDOS for each element. The vacuum energy level $(E_{vacuum})$ is shown in dotted lines in PDOS. Individual atomic surface structures are shown in Supplementary Fig. 25. The Fermi level $(E_{Fermi})$, which is calculated from DFT calculations and represents the highest energy level that is occupied by electrons at zero K temperature, is shown in dashed lines in PDOS. Enlarged plots of the PDOS around the VBM of defective CFPB are shown to present the unoccupied states (mostly from $Fe^{3+}$) due to the Co vacancy. **b** Band alignments between different CFPB based on the vacuum level and the corresponding Standard Hydrogen Electrode (SHE) potential, where

$E_{abs} = E_{SHE} + 4.44$ V. The shaded regions represent the bands that are occupied with electrons, green for Fe and blue for Co contributions (obtained from PDOS). The empty regions around the VBM of non-stoichiometric Co deficient and defective CFPB mean the unoccupied bands that are capable of accepting electrons. The reduction potential of water (1.23 V vs. SHE), oxygen (0.7 V vs. SHE), and $H_2O_2$ to form hydroxyl radical (0.3 V vs. SHE), are shown in the dash-dotted line, dotted line, and dashed line, respectively. **c** Illustration shows the mechanism using the active sites-isolated CFPB catalyst for the tandem reaction (water oxidation reaction, ORR, and Fenton-like reaction). The innate feature of inter-charge transfer in CFPB can provide aid in renewing the electron configuration of different active sites after the catalysis reactions.

---

of the Co vacancy that results in some valence bands being unoccupied, indicating some of the $Fe^{2+}$ converted to $Fe^{3+}$, in line with our FTIR results shown in Fig. 2c. Figure 4b shows the band alignments of different CFPB with respect to the vacuum level. The non-stoichiometric (Fe-rich) and Co- and Fe-defective CFPB show a work function >6 V, reducing the unoccupied valence bands below the water oxidation potential, implying a possible driving force for electron transfer from water to $Fe^{3+}$ and production of $O_2$. On the other hand, the Fermi level (occupied states) of the non-stoichiometric (Co-rich) CFPB is higher than the reduction potential of $O_2$ and $H_2O_2$, indicating a potential electron transfer from the CFPB to them, forming the $H_2O_2$ and OH radical. Therefore, the key factor that enables CFPB to multifunctionally activate the $H_2O$ oxidation, oxygen reduction, and Fenton-like reaction is the CFPB structure composed of mutually isolated sites with different crystal phases and compositions (Fig. 4c). Without isolated characteristics, the electron transfer would occur within the CFPB instead of to/from the surrounding molecules.

## In vitro evaluation of CDT and PTT and cytotoxicity studies

HepG2-Red-Fluc (human hepatocellular carcinoma cell line with expression of luciferase) and A549 cells (lung carcinoma cells) were chosen for proof of concept through in vitro evaluation of CDT and PTT for CFPB nanocubes and nanoframes. The ·OH levels in cell cultures are evaluated using APF. Time- and dose-dependent ·OH generation was seen from CFPB nanoframes, CFPB@Lipo nanoframes, and CFPB@Lipo nanocubes (Fig. 5a). Notably, no immediate ·OH production from CFPB@Lipo nanostructures was observed in the first 10 min of incubation, indicating an early-stage silent water-driven chemodynamic reaction. Significant ·OH production was observed as the incubation time increased from 4 to 20 h for CFPB@Lipo nanoframes, suggesting that the lipid membrane covering the nanoframe surface fuses with the cell plasma membrane, thus the exposed CFPB nanoframes form water-driven ·OH. Cell images for intracellular $O_2$ $([Ru(dpp)_3]Cl_2$ used as an indicator with red fluorescence in the absence of $O_2$), $H_2O_2$ (2′,7′-Dichlorodihydrofluorescein diacetate (DCFH-DA) used as an indicator with green fluorescence) and ·OH (APF used as an indicator with green fluorescence) all reveal stronger fluorescent behavior in CFPB@Lipo nanoframes (Fig. 5b–d). Higher intracellular $H_2O_2$ and ·OH generation can be attributed to the higher cellular uptake of the Lipo coating when CFPB@Lipo nanoframes are compared to CFPB nanoframes without Lipo (Supplementary Fig. 10). However, we cannot exclude the possibility that endocytosis might be inhibited by premature cell membrane damage because of uncontrollable ·OH formation from the CFPB nanoframes, resulting in a relatively low cellular uptake of CFPB nanoframes in the absence of the Lipo covering. Flow cytometry analysis demonstrates significantly increased early- and late- apoptotic cells after 24 h incubation with CFPB@Lipo nanoframes (Fig. 5e). The cells incubated with CFPB@Lipo nanoframes for 24 h have a relatively higher late apoptotic ratio (35.57%) compared to other groups (cell only: 0.06%, CFPB nanoframes: 1.25% and CFPB@Lipo nanocubes: 12.48%). Live and dead staining experiments reveal consistent results, indicating that this

water-driven chemodynamic reaction by CFPB@Lipo nanoframes can be facilitated by the excellent cell affinity from the exposed lipid membrane (Fig. 5f). Similar results were shown in the A549 lung carcinoma cells line (Supplementary Fig. 11).

Next, the cell survival rate was examined as a function of CFPB@Lipo nanoframe dosage following CDT alone (without laser exposure) and CDT + PTT (with laser exposure) (Supplementary Fig. 12a). A dosage-dependent profile was observed but discrepancies appeared only when the nanoframe concentrations increased to 28 ppm for the observation of the photothermal effect. This is because the heating temperature reached 42℃ in the cell culture system after nanoframe concentrations reached 28 ppm (Supplementary Fig. 12b). While previous studies have reported that the thermal effect can enhance the chemodynamic effect[44], no such result was observed in the resulting nanoframes (Supplementary Fig. 13). Finally, for in vivo animal studies, we must prevent the generation of additional ·OH in blood circulation. CFPB and CFPB@Lipo nanoframes were individually incubated with blood containing 2% red blood cells to evaluate ·OH production. No appreciable sign of ·OH was observed from CFPB@Lipo across the dosages while an enhanced signal was detected at higher dosages from the CFPB nanoframes (Supplementary Fig. 14a). No hemolytic phenomenon or vascular endothelial cell damage was seen from the CFPB@Lipo nanoframes (Supplementary Fig. 14b, c).

## Treatments of orthotopic and superficial in vivo tumors

Rhodamine (RB) was encapsulated into NPs to facilitate ex vivo imaging observation through bright fluorescence (ex.: 535 nm; em.: 580 nm). Considering that CFPB nanocubes are also capable of ·OH production (Fig. 3), RB-loaded CFPB@Lipo nanocubes and nanoframes were prepared and administrated via tail vein for investigation. Predominant accumulation in the liver and kidneys at 3 h post injection and subsequent fluorescence disappearance post-24 h injection which may be due to metabolism issues (Supplementary Fig. 15a). In vivo biodistribution determined by atomic absorption (AA) measurements also shows the same trend. The results are consistent with ex vivo imaging showing the highest NP accumulation in the liver and kidneys at 3 h post-injection and significant elimination after 24 h (Supplementary Fig. 15b).

For the orthotopic tumoral studies, we established a HepG2-Red-FLuc based hepatocellular carcinoma animal model for bioluminescence to monitor tumor growth. Mice were divided into three treatment-based cohorts: PBS, CFPB@Lipo nanocubes, and CFPB@Lipo nanoframes. A single dose (100 ppm/mouse) was intravenously injected into the mice and tumor sizes were observed using IVIS (Supplementary Fig. 16a–c). PBS group tumors grew continuously, while both the CFPB@Lipo nanocubes and CFPB@Lipo nanoframes groups saw reduced tumor growth rates. Tumor growth inhibition was more pronounced in the CFPB@Lipo nanoframes group than in the CFPB@Lipo nanocubes group (Supplementary Fig. 16d). However, the tumors could not be suppressed efficiently and grew over the course of the day for both CFPB@Lipo nanocubes and CFPB@Lipo

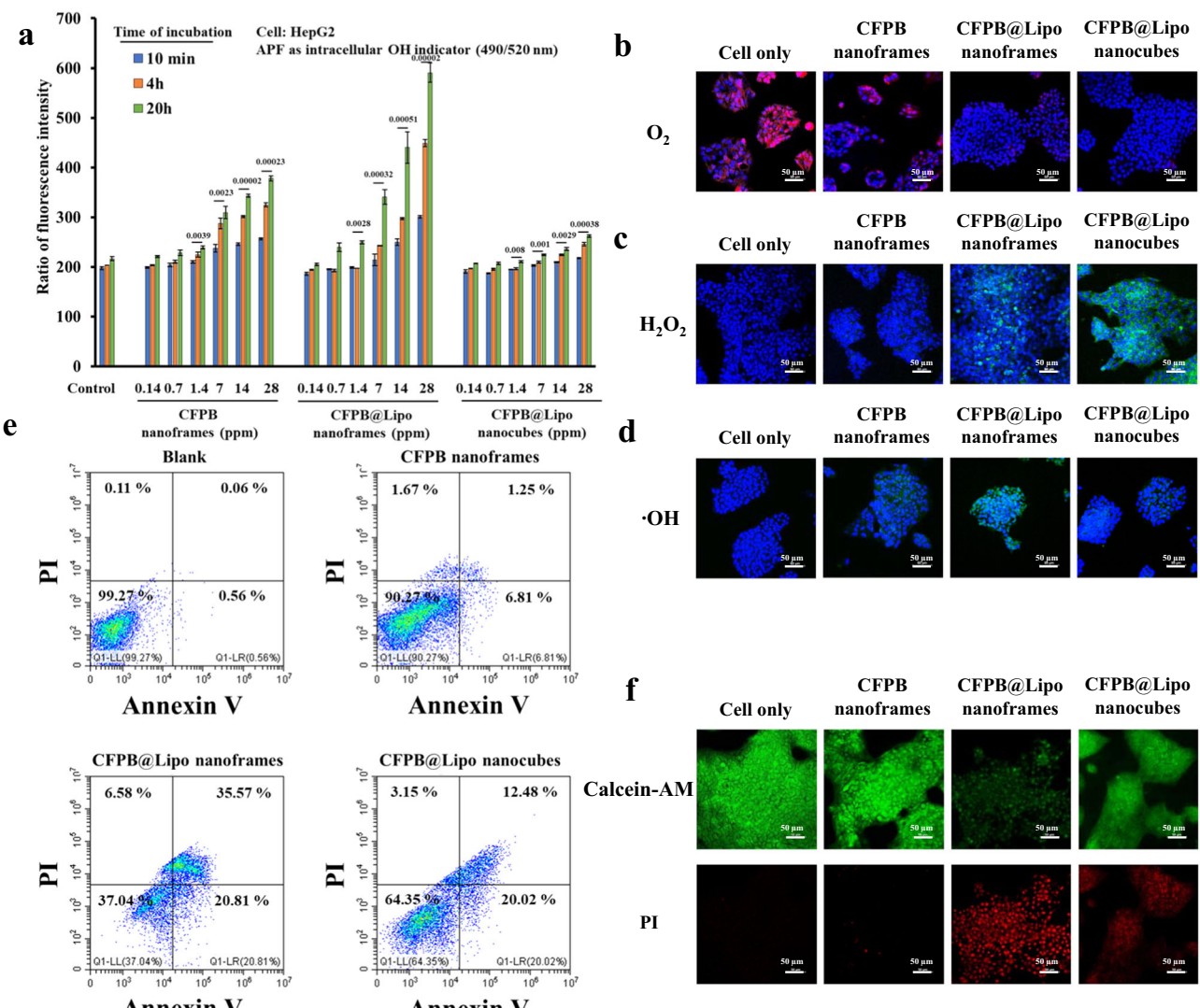

**Fig. 5 | In vitro studies of CDT and cytotoxicity studies. a** Fluorescence intensity of APF representing the level of ·OH generated from the control group (buffer only), CFPB nanoframes, CFPB@Lipo nanoframes and CFPB@Lipo nanocubes under various concentrations over 10 min, 4 h, and 20 h. **b** HepG2-Red-FLuc cells treated with [Ru(dpp)$_3$]Cl$_2$ dye (as the control group), [Ru(dpp)$_3$]Cl$_2$ dye + CFPB nanoframes, [Ru(dpp)$_3$]Cl$_2$ dye + CFPB@Lipo nanoframes and [Ru(dpp)$_3$]Cl$_2$ dye + CFPB@Lipo nanocubes under a 30-min incubation to monitor O$_2$ generation. **c** HepG2-Red-FLuc cells treated with DCFH-DA dye (as the control group), DCFH-DA dye + CFPB nanoframes, DCFH-DA dye + CFPB@Lipo nanoframes, and DCFH-DA dye + CFPB@Lipo nanocubes under a 30-min incubation to monitor H$_2$O$_2$ generation (green emissions). **d** HepG2-Red-FLuc cells treated with APF dye (as the control group), APF dye + CFPB nanoframes, APF dye + CFPB@Lipo nanoframes, and APF dye + CFPB@Lipo nanocubes under a 30-min incubation to monitor H$_2$O$_2$ generation (green emissions). **e** Flow cytometry analysis of HepG2-Red-FLuc cancer cells with and without CFPB nanoframes, CFPB@Lipo nanoframes, and CFPB@Lipo nanocubes. **f** Live and dead staining for CFPB nanoframes, CFPB@Lipo nanoframes, and CFPB@Lipo nanocubes. Significant damage in cells is seen in CFPB@Lipo nanoframes. All scale bars are 50 μm. All data were obtained in triplicate ($n = 3$, the error bars represented mean ± SD). $p$-values were calculated by one-way ANOVA. Source data are provided as a Source Data file. **b**–**d**, **f**, One representative data was shown from three independently repeated experiments.

nanoframes (Supplementary Fig. 16c). Because of tumor recurrence following the single dose, two dose treatments were then arranged (with the 2nd dose administrated on the post-7th day). The group treated with CFPB@Lipo nanoframes showed promising tumor growth inhibition (Fig. 6a–c and Supplementary Fig. 17). The TGI profile of CFPB@Lipo nanoframes group showed steady tumor suppression of ~97% after 18 days of observation (Fig. 6d). No significant change in body weight was observed for either single or two dose treatments (Fig. 6e and Supplementary Fig. 16e). Both the CFPB@Lipo nanocubes and CFPB@Lipo nanoframes treated groups displayed DNA damage (Fig. 6f and Supplementary Fig. 16f). The O$_2$ generation due to the water oxidation reaction in the nanocubes and nanoframes also resulted in tumor hypoxia relief following HIF-1α antibody assay (Fig. 6f).

An 808 nm NIR diode laser cannot effectively reach orthotopic hepatic tumors because of penetration limitations. Thus, a therapeutic examination was performed for PTT using a superficial tumor model to evaluate the photothermal capability of CFPB nanoframes. The superficial tumor model was established using an A549 (human alveolar basal epithelial cancer cell line) animal model. As seen in Fig. 6g, the CFPB@Lipo nanoframes + laser (808 nm exposure at 0.8 W/cm$^2$ for 10 min) suppressed the tumor within 1 day with no tumor recurrence over time, indicating that the CFPB@Lipo nanoframes are applicable as a photothermal agent (Fig. 6g and Supplementary Fig. 18). Finally, biochemical blood test results show no significant difference in liver and kidney function indexes (Supplementary Fig. 19) and no obvious histological damage within normal organs (Supplementary Fig. 20) for the CFPB@Lipo

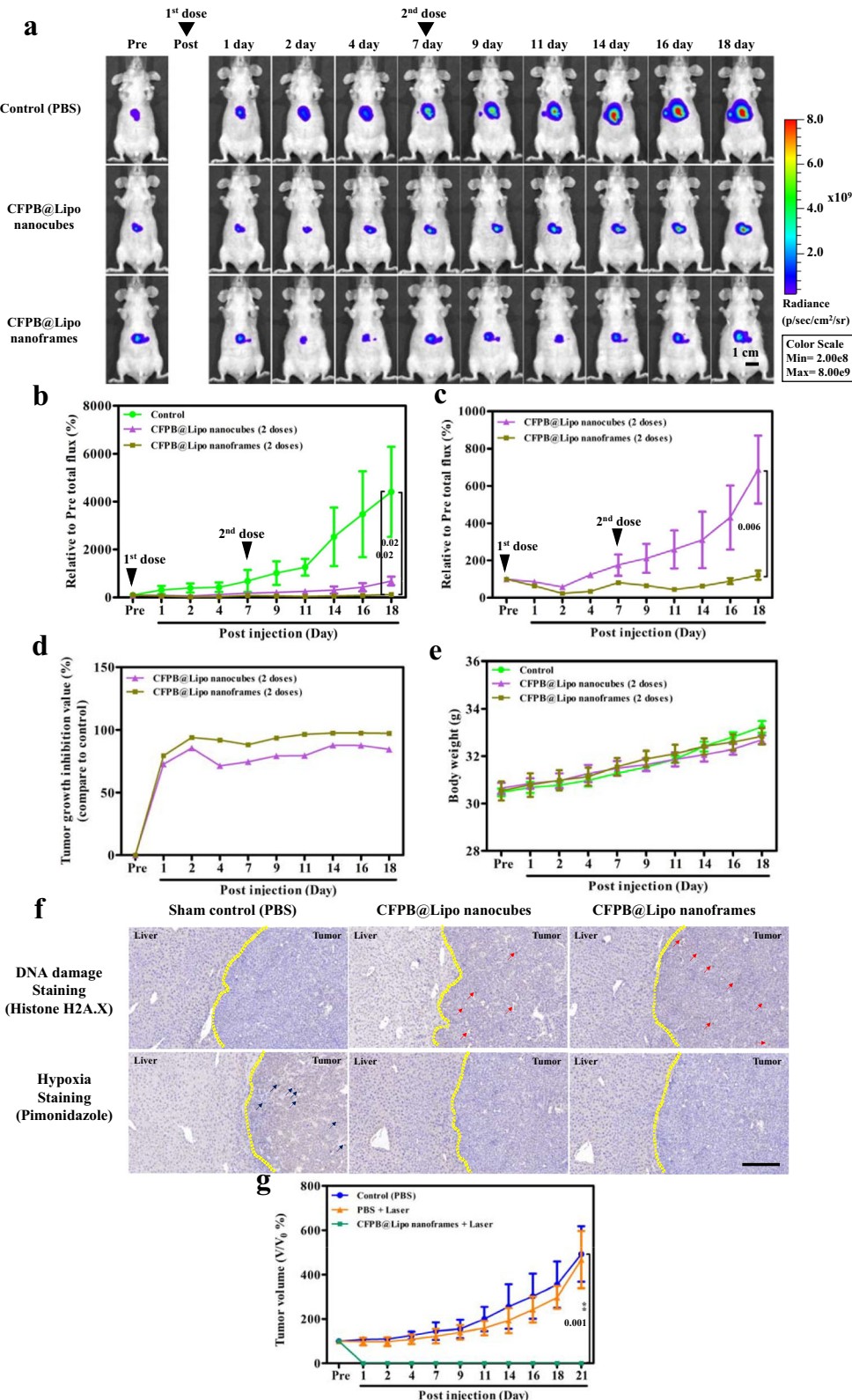

**Fig. 6 | In vivo anti-tumor activity of the mice with Hep G2-Red-FLuc orthotopic tumors following 2 doses administration and superficial model for laser exposure. a** The animal bioluminescence images from Hep G2-Red-FLuc cells were monitored using the IVIS imaging system. The mice were individually intravenously injected two doses (one dosage: 100 ppm/mouse) of CFPB@Lipo nanocubes and CFPB@Lipo nanoframes and the control set was injected with PBS. The 2nd dose was administrated on post 7th day. **b** The tumor growth profiles from the different treated groups. **c** The tumor growth profiles without control group from **b**. **d** Illustration of the tumor growth inhibition (TGI) rate. **e** The variation of body weight from the different treated groups. **f** The DNA damage and hypoxia observation of the tumor region in liver tissue. The expression of phospho-H2A.X and pimonidazole were detected by IHC staining (red arrows: DNA damage markers within liver tumor cells; blue arrows: hypoxia staining, scale bar: 200 μm). **g** Superficial tumor growth curves with diode laser exposure at 0.8 W/cm² for 10 min. All data were obtained in triplicate (n = 3). The error bars represented mean ± SD, p-values were calculated by one-way ANOVA. Source data are provided as a Source Data file. **a**, **f**, One representative data was shown from three independently repeated experiments.

nanoframes-treated mice. CFPB@Lipo nanocubes reveal no toxicity as well.

## Discussion

Figures 1–2 show unexpected results indicative of compositional changes in CFPB during acid corrosion. The examinations are extended to other Prussian blue analogs, i.e., MnFe Prussian blue (MFPB) and NiFe Prussian blue (NFPB), following this acid-etching reaction (see preparation in Supplementary). MFPB is found to result in dissolution fragments and NFPB forms a frame-like structure (Supplementary Fig. 21). Notably, this acid corrosion for both MFPB and NFPB all result in the same NIR absorption seen in CFPB. However, the PB cubes are quite resistant to acidic corrosion (Supplementary Fig. 22).

A proton-induced metal replacement reaction is proposed for the etching process (Fig. 2g). The acidic corrosion of all PBAs (i.e. CFPB, MFPB, and NFPB) involves the proton-induced metal replacement reactions enabling NIR absorption $ex$ $nihilo$. This corrosion is associated with two reactions: acid etching and metal ion replacement. In acid etching in CFPB, the $Co^{2+}-N\equiv C-Fe^{3+}$, $Co^{2+}-N\equiv C-Fe^{2+}$, and $Co^{3+}-N\equiv C-Fe^{2+}$ compositions of the CFPB nanocubes are decomposed to yield various fragments, i.e., $Co^{2+}$, $Co^{3+}$, $[Fe(CN)_6]^{3-}$, and $[Fe(CN)_6]^{4-}$. Both $[Fe(CN)_6]^{3-}$ and $[Fe(CN)_6]^{4-}$ react with protons to produce $Fe^{2+}$, $Fe^{3+}$, and HCN[45]. The $Fe^{2+}$ ions are further oxidized to $Fe^{3+}$ in an oxygen-containing environment[46]. The generated $Fe^{3+}$ substitutes the cobalt ions in the unetched compositions of $Co^{3+}-N\equiv C-Fe^{2+}$ and $Co^{2+}-N\equiv C-Fe^{2+}$ to form $Fe^{3+}-N\equiv C-Fe^{2+}$, causing increased NIR absorption in the frame structure. The $Co^{2+}/Co^{3+}$ ratios in the nanostructures through XPS analysis increase gradually from 1.15 to 1.93 as the reaction progresses, indicating that the metal substitution favors the replacement of $Co^{3+}$ in $Co^{3+}-N\equiv C-Fe^{2+}$ by $Fe^{3+}$ relative to $Co^{2+}$ in $Co^{2+}-N\equiv C-Fe^{2+}$ (Fig. 2f). The ionic radius of $Fe^{3+}$ is 0.064 nm, closer to that of $Co^{3+}$ (0.063 nm) than $Co^{2+}$ (0.074 nm). This supports the favorable substitution reaction between $Fe^{3+}$ and $Co^{3+}-N\equiv C-Fe^{2+}$. In addition, the released $Fe^{2+}$ should also have a chance to replace $Co^{2+}$ of $Co^{2+}-N\equiv C-Fe^{3+}$, resulting in $Fe^{2+}-N\equiv C-Fe^{3+}$ followed by charge transfer to yield $Fe^{3+}-N\equiv C-Fe^{2+}$.

CFPB-catalyzed water oxidation reaction and ORR require external energy inputs[30–34,36]. The present study is the first to demonstrate the excellent performance of CFPB in the generation of $O_2$, $H_2O_2$, and $\cdot OH$ from $H_2O$ without stimulus. The water oxidation reaction or oxygen evolution includes four main steps: (1) water association with the catalytic metal site (M), (2) oxidizing water to form M = O, (3) reacting with another water molecule for O-O bond formation, and (4) M releasing oxygen to expose the vacancy for the following reaction cycle[47]. In oxygen evolution, the oxidation activation of the catalyst is the most critical factor. Experimental and simulation results demonstrate that $Fe^{3+}$, as the active site in $Co^{2+}-N\equiv C-Fe^{3+}$ in the Fe rich non-stoichiometric, Fe defective and in Co defective crystal phases, shows high reduction potentials of respectively 2.07, 2.12, and 1.84 V to initiate water oxidation, producing $O_2$ and $Co^{2+}-N\equiv C-Fe^{2+}$. Like the water oxidation reaction, ORR also requires a suitable catalyst. The process includes the oxygen absorption by the metal site of the catalyst, followed by two $H^+$/electron couples involving O-H bonds formation to generate $H_2O_2$[48]. According to our experimental and simulation results, ORR can then be driven by $Co^{2+}$ (highest oxidation potential $E = -0.25$) in $Co^{2+}-N\equiv C-Fe^{2+}$ in the Co-rich non-stoichiometric crystal phase to produce $H_2O_2$ and $Co^{3+}-N\equiv C-Fe^{2+}$. The high spin $Co^{2+}$ abundant in CFPB serves as the catalytic centers to trigger a chemodynamic reaction[39–49]. The inter-charge transfer in $Co^{3+}-N\equiv C-Fe^{2+}$ returns the composition to $Co^{2+}-N\equiv C-Fe^{3+}$, thus initiating another run of catalysis reaction. In this process, CFPB functions as an active site-isolated catalyst with the distinct ability to facilitate the sustainable conversion of $H_2O$ to $H_2O_2$. The compositions and nanostructure of CFPB nanocubes (with more $Fe^{3+}$) and CFPB nanoframes (with more $Co^{2+}$) lead to differentiation in the production of $O_2$ and $\cdot OH$.

In summary, an intriguing reaction through proton-induced metal replacement is found to convert CFPB nanocubes with no NIR absorption into CFPB nanoframes with NIR absorption due to an increased $Fe^{3+}-N\equiv C-Fe^{2+}$ composition. Other PB analogs (i.e., MFPB and NFPB) also follow the same process. Importantly, both CFPB nanocubes and nanoframes can produce water oxidation, oxygen reduction and Fenton-like reactions. CFPB nanoframes have greater capability to generate sustainable $\cdot OH$ production for a water-driven CDT.

## Methods

### Statement

Animal care was provided in accordance with the Laboratory Animal Welfare Act and the Guide for the Care and Use of Laboratory Animals and approved by the Institutional Animal Care and Use Committee of Kaohsiung Chang Gung Memorial Hospital (KCGMH). All animal treatments and surgical procedures were performed in accordance with the guidelines of KCGMH Laboratory Animal Center (IACUC NO. 2021031802).

### Chemicals

All regents were analytical purity and used without further purification. Potassium hexacyanoferrate(III) ($K_3[Fe(CN)_6]$, 99%) was bought from Riedel-de-Haën. Trisodium citrate dihydrate ($C_6H_5Na_3O_7\cdot2H_2O$, 99%) was obtained from SHOWA. Hydrochloric acid (HCl, 36%) was bought from BASF. Manganese(II) acetate ($C_4H_6MnO_4$, 98%), and tris(4,7-diphenyl-1,10-phenanthroline)ruthenium(II) dichloride complex ($Ru(ddp)_3Cl_2$, $C_{72}H_{48}Cl_2N_6Ru$) were obtained from Alfa Aesar. Ethanol ($C_2H_5OH$, 99.9%) was purchased from J. T. Baker. Aminophenyl fluorescein solution (APF, $C_{26}H_{17}NO_5$, 98%) was acquired from Life Technologies. Cobalt(II) acetate ($C_4H_6CoO_4$, 99%), Nickel(II) acetate tetrahydrate ($C_4H_6NiO_4\cdot4H_2O$, 98%), Polyvinylpyrrolidone (PVP, $(C_6H_9NO)n$, M.W. = 55,000), hydrogen peroxide solution ($H_2O_2$, 30%), 1,2-dioleoyl-sn-glycero-3-phosphocholine (DOPC, $C_{44}H_{84}NO_8P$, 99%), 1,2-dioleoyl-sn-glycero-3-phosphoethanolamine (DOPE, $C_{41}H_{78}NO_8P$, 99%), potassium iodide (KI, 99.5%), and 3-(4,5-dimethylthiazol-2-yl)-2,5-diphenyltetrazolium bromide (MTT, $C_{18}H_{16}BrN_5S$, 97.5%) were bought from Sigma-Aldrich. Water was obtained by using a Millipore direct-Q deionized water system throughout all studies.

### Synthesis of cobalt-iron Prussian blue (CFPB) nanocubes

0.05 g of potassium hexacyanoferrate(III) was dissolved in 30 mL of distilled deionized water (DDW). 0.075 g of cobalt(II) acetate and 0.147 g of trisodium citrate dihydrate were dissolved in another 20 mL of DDW. Then, both solutions were mixed together and stirred in ice bath. After 15 min, the precipitates were collected through centrifugation at 8700 g for 5 min, and washed with 50%/50% of ethanol/deionized water (volume/volume). We repeatedly washed and centrifuged the CFPB nanocubes at least three times. Finally, the CFPB nanocubes were dispersed in deionized water for future use. The size of the CFPB was analyzed using the SigmaScan Pro 5 software (version 5.0.0).

### Acid-etching reaction of CFPB nanocubes

CFPB nanocubes (400 ppm in Fe ion concentration) were dissolved in 2 mL of DDW, and then mixed with 14 mL of the acidic solutions (0.01 M HCl). Next, the solution was stirred and heated in oil bath at 90°C for different times (0.5, 3, 6, 16 and 24 h). After reaction, the solution was collected, and then centrifuged at 7800 g for 5 min. The supernatants were removed and the precipitates were redispersed into 50%/50% of ethanol/deionized water (volume/volume). We repeatedly washed and centrifuged the nanoframes for at least three times.

Finally, the nanoframes were dispersed in deionized water for future use.

## Preparation of liposomes

Liposomes were prepared using the extrusion method with an Avanti Polar Lipids, Inc. extruder. DOPC and DOPE were chosen for their structural similarity to natural cell membranes, enabling successful fusion with them. The technique of formulating DOPC/DOPE-combined liposomes with this capability is well-established[50]. For this study, liposomes were prepared with a fixed ratio of 4 parts DOPC to 1 part DOPE. Initially, 100 mg of DOPC and DOPE powders were separately dispersed in 5 mL of chloroform to create lipid stock solutions. A mixture containing 0.252 mL of DOPC and 0.059 mL of DOPE was dried using $N_2$ gas purging to form a multi-layered lipid film. The film was then dispersed in 0.8 mL of PBS solution to obtain a lipid work solution. Subsequently, the lipid work solution was extruded through a polycarbonate membrane with 100 nm pore size using the extruder, employing forward and reverse extrusions for at least five cycles. This process yielded monodisperse 100 nm liposomes composed of 80% DOPC and 20% DOPE. The resulting liposomes were stored in a 4 °C refrigerator for future use.

## Preparation of CFPB@Lipo nanoframes

18 mg liposomes were added in the 1 mL of CFPB nanoframes (125 ppm cobalt ion concentration) dispersed in PBS. For the preparation of CFPB@Lipo-RB nanoframes, the additional 0.1 mg of rhodamine B was added into the above solution. Afterward, the mixture was treated by sonication at room temperature for 15 min. And then, the mixture was placed in the cooling box for 5 min. Through applying three cycles of ultrasound-cooling process, the well-coating of lipid membrane on nanoframes was achieved. CFPB@Lipo or CFPB@Lipo-RB nanoframes were then washed by PBS at least three times, and stored in 4 °C for future use.

## Evaluation of morphology using electron microscopy

The morphology of CFPB nanocubes and CFPB nanoframes was evaluated using both transmission electron microscopy (Hitachi H-7500) and scanning electron microscopy. During the experiments, solutions containing CFPB nanocubes or CFPB nanoframes (1 ppm) were dropped onto copper grids. After the solvent dried, the copper grids were delivered to the electron microscopy for morphology measurement.

## Evaluation of zeta potential

The zeta potential of Lipo, CFPB nanoframes, and CFPB@Lipo nanoframes was evaluated using a zeta potential analyzer (Malvern, Zetasizer nano ZS90). During the experiments, solutions containing 1 ppm of nanoparticles were dispersed in PBS. The solutions were then aspirated into cells for zeta potential analysis.

## Evaluation of surface functional groups

The functional groups of Lipo and CFPB@Lipo nanoframes were evaluated using a Fourier Transform Infrared Spectrometer (Perkin Elmer). During the experiments, the Lipo and CFPB@Lipo were dried into powder. Then, 0.02 g of the sample was ground with 0.1 g of KBr. After pressing into a plate, it was delivered to the Fourier Transform Infrared Spectrometer for measurement.

## Evaluation of weight loss

The weight loss of CFPB nanoframes and CFPB@Lipo nanoframes was evaluated using Thermogravimetric analysis (PISC015). During the experiments, the CFPB nanoframes and CFPB@Lipo nanoframes were dried into powder. Then, 0.2 g of the sample was sent to Feng Chia University, with a setting temperature of 700 °C and a temperature increase rate of 0.2 °C/min

## Evaluation of oxygen generation

Prior to $O_2$ measurement, $O_2$ is removed from the test tube using a purge procedure with $N_2$. The $Ru(dpp)_3Cl_2$ is used as an indicator for the generation of oxygen. The solution containing 100 µL of $Ru(dpp)_3Cl_2$ (40 µM) + 0.9 mL of $H_2O$ was prepared as a control group. $Ru(dpp)_3Cl_2$ (40 µM) was added to 0.9 mL of CFPB nanocubes and nanoframes individually at different concentration (0.14, 0.7, 1.4, 7 and 14 ppm in cobalt ion concentration) at 37 °C under dark for 1, 3, 5, 10, 15 and 30 min. After reaction, the samples were centrifuged at 8.7 g for 5 min to obtain $Ru(dpp)_3Cl_2$ supernatants. The relative ratio of fluorescence intensity was obtained from $Ru(dpp)_3Cl_2$ supernatant (ex/em: 455/613 nm) to reflect the presence of oxygen. All of the data were obtained in triplicate. The statistical analysis was calculated by Origin 9.

## Quantitation of $H_2O_2$ generation by colorimetric test

The $H_2O_2$ solution reacts with potassium iodide (KI) to generate yellow $I_3^-$ and then quantified by UV−Vis measurements. The reactions are shown below:

$$KI + H_2O_2 \rightarrow I_2 + 2KOH$$

$$KI \rightarrow K^+ + I^-$$

$$I_2 + I^- \rightarrow I_3^-$$

For the standard calibration curve preparation, the 1 mL solutions containing 0.5 M KI (excess) with different concentrations of $H_2O_2$ (1, 2, 5, 10, 20, 50 and 100 µM) for 10 min reaction produced $I_3^-$. The calibration curve was obtained from 350 nm absorbance of $I_3^-$ vs corresponding concentration of $H_2O_2$.

The 0.9 mL solution containing 14 ppm of CFPB nanocubes or nanoframes was sealed and purged by $N_2$ for 10 min to remove the oxygen from vial. The 100 µL of KI (0.1 M) purged by $N_2$ was added in the vials of CFPB nanocubes or nanoframes for different reaction times (1, 3, 5, 10, 15 and 30 min) under dark, resulting in a solution color changed to yellow due to $H_2O_2$ production. After reaction, the solutions were centrifuged at 8700 g for 5 min, and then the yellow supernatants were measured to obtain the absorbance of $I_3^-$, which could be further analyzed through the standard calibration curve for quantitation of hydrogen peroxides. For the evaluation of the different concentrations, the 0.9 mL solutions containing CFPB nanocubes or nanoframes (0.14, 0.7, 1.4, 7 and 14 ppm in cobalt ion concentration) were analyzed following the same processes (reaction is 30 min) to obtain correlated absorbance of $I_3^-$.

## Evaluation of hydroxyl free radicals generation

The APF dye is used as an indicator to determine the generation of hydroxyl free radicals. The 0.1 mL PBS solution containing 10 µM of APF was added in the each wells of 96-well plates. And then, the colloidal CFPB nanocubes, CFPB nanoframes or CFPB@Lipo nanoframes at different concentrations (0, 0.14, 0.7, 1.4, 7, 14 and 28 ppm in cobalt ion concentration) were added in the wells containing APF for the reaction at dark. If the additional photothermal treatment was conducted, the solutions were irradiated with 808 nm diode laser at 0.8 W/cm² power density for 10 min. Afterward, the the fluorescence intensity of APF dye (ex/em: 490/520 nm) in individual wells was measured by ELISA reader (BioTek Synergy HTX) at the individual reaction time points (1, 3, 5, 10, 15 and 30 min). The relative ratio of fluorescence intensity was calculated from the fluorescence intensity of APF dye + nanomaterials measured at different reaction times divided by fluorescence intensity of pure APF dye. All of the data were measured in triplicate.

For the evaluation of ·OH in the cells, the cancer cells were seeded in 96-well plates (8000 HepG2-Red-FLuc cells/per well) and incubated

for 24 h, followed by adding APF dye with a finial concentration of 10 μM. After 10 min incubation, different concentrations of CFPB or CFPB@Lipo nanoframes (0, 0.14, 0.7, 1.4, 7, 14 and 28 ppm in cobalt ion concentration) were added with the cells for further incubation of 10 min, 4 h and 20 h. After incubation, the fluorescence intensity of APF in the well was directly measured by ELISA reader. All of the data were obtained in triplicate.

For ·OH evaluation in blood, the solutions containing 10 μM of APF, 2% red blood cells, and CFPB or CFPB@Lipo nanoframes at different concentration (0, 0.14, 0.7, 1.4, 7 and 14 ppm in cobalt ion concentration) were stood in dark at 37 °C for 5 min. The fluorescence intensity at 520 nm was measured by the fluorescence spectrophotometer, and the relative ratio of fluorescence intensity was calculated to evaluate ·OH level generated in blood. All of the data were measured in triplicate. The statistical analysis was calculated by Origin 9.

### Evaluation of hydroxyl free radicals by ESR
ESR analysis was employed to assess the generation of ·OH, utilizing DMPO spin-trapping adduct. In the experimental setup, PBS-dispersed solutions containing 100 mM DMPO and either CFPB nanocubes or CFPB nanoframes were prepared. Subsequently, these mixtures were aspirated into quartz capillaries for the ESR evaluation.

### Cellualr uptake of CFPB and CFPB@Lipo nanoframes
$0.8 \times 10^4$ HepG2-Red-FLuc cells/well were grown in 96-well plates for 24 h. The 50 ppm (in cobalt ion concentration) of CFPB or CFPB@Lipo nanoframes were added in the wells for 4 h incubation at 37 °C. After incubation, the old mediums containing residual CFPB or CFPB@Lipo nanoframes were removed and the cells were washed by fresh PBS at least three times. After wahsing, the cells were immersed in aqua regia for 1 day. Finally, the cobalt ion concentrations of aqua regia were determined by atomic absorption (AA) measurements. The statistical analysis was calculated by Origin 9.

### Stability of CFPB@Lipo nanoframes
The CFPB@Lipo nanoframes were dispersed in water, PBS (pH7), PBS (pH5), and 10% serum in dark at 37 °C for 7 days. The TEM was used for analysis of their structure stability.

### Cell culture
A549 (human alveolar basal epithelial cancer cell line, 60074, BCRC) cells were cultured in DMEM containing NEAA (0.1 mM), penicillin/streptomycin (PS, 1%), and fetal bovine serum (FBS, 10%) in the incubator at 37 °C and 5% CO₂. Hep G2-Red-Fluc (Human liver cancer cell line, BW134280V, PerkinElmer) cells were maintained in high glucose DMEM (Caisson, Smithfield, VA, USA) supplemented with 10% FBS and the antibiotics penicillin/streptomycin All cells were maintained at 37 °C in a humidified atmosphere containing 5% CO₂. HUV-EC-C cells (endothelial cell line, 63872098, ATCC) were cultured in F-12k containing EGCS (0.03 mg/mL), heparin (0.1 mg/mL), and fetal bovine serum (FBS, 10%) in the incubator at 37 °C and 5% CO₂.

### Flow cytometry assay
A549 and HepG2-Red-FLuc cells were seeded in a 6-cm culture dish with a population of $5 \times 10^5$ cells and incubated overnight. Cells were then treated with CFPB nanoframes or CFPB@Lipo nanoframes at 28 ppm (in cobalt ion concentration). Control cells (culture medium only as negative control and 2 μM Thapsigargin as positive control) were also included in this experiment. After 24 h, cells were washed twice with PBS and were later detached by trypsinisation. Then, the cells were harvested and washed with cold PBS. Cells were then re-suspended in 500 μL of 1X annexin V binding buffer. Next, 2 μL of annexin-V and 1 μL of PI (propidium iodide) were added to cells. Cells were gently vortexed, incubated at room temperature for 15 min and analyzed by flow cytometry (CytoFLEX S, Beckman Coulter (version

2.5.0.77)). Cell populations were gated with forward scatter (FSC) and side scatter (SSC) plot of cell only. The gate was set to remove dead cells and aggregates cells. (Supplementary Fig. 23).

### Heating performance of nanomaterials upon laser Irradiation
The change of temperature in solution was recorded by a digital thermometer (TES 1319A-K type). For the evaluation of heating performance, the 100 μL of water, CFPB nanocubes or CFPB nanoframes (14 ppm in cobalt ion concentration) were exposed to an 808 nm diode laser at 0.8 W/cm² power density. For the evaluation of photothermal effect in cell culture, the A549 cells (8000 cells/per well) were treated with different concentrations of CFPB@Lipo nanoframes (14, 28 and 56 ppm in cobalt ion concentration) for 4 h incubation, and then the cultures were irradiated by an 808 nm diode laser at 0.8 W/cm² for 10 min.

### Cells viability following laser exposure
The cancer cells were seeded in 96-well plates (8000 A549 cells/per well) and incubated for 24 h. The medium was removed, and then fresh medium was added to the culture with different concentrations of CFPB@Lipo nanoframes (0, 0.14, 0.7, 1.4, 7, 14, 28 and 56 ppm in cobalt ion concentration). The cells were incubated for another 24 h, followed by a wash with fresh medium for at least 3 times for the groups without laser expsore. For the group with additional photothermal treatment, the cells were irradiated with 808 nm diode laser at 0.8 W/cm² power density for 10 min after 4 h incubation. After laser irradiation, the cells were incubated for additional 20 h, followed by a wash with fresh medium for at least 3 times. Finally, the medium of 10% MTT reagent was used to added in the wells, and then the cells were incubated for 4 h. The standard ELISA method was applied to determine the cell viability. All data was obtained in triplicate.

### Hemolysis analysis
The 2% red blood cells were prepared in deionized water (positive control group) and PBS containing with CFPB@Lipo nanoframes at different concentration (0, 0.14, 0.7, 1.4, 7, 14 and 28 in cobalt ion concentration). These solutions were stood at dark for 1 h. And then, the solutions were centrifuged at 8.7 g for 5 min to evaluate the hemolysis condition.

### In vitro evaluation by fluorescence imaging
2′,7′-Dichlorodihydrofluorescein diacetate (DCFH-DA) and APF dyes were utilized to sense the intracellular $H_2O_2$ and ·OH radicals, respectively. The cancer cells were seeded in 96-well plates (8000 HepG2-Red-FLuc and A549 cells/per well) and incubated for 24 h, followed by the treatments of DCFH-DA or APF dyes with the CFPB and CFPB@Lipo nanoframes (28 ppm in cobalt ion concentration) for another 24 h. The cells were treated by the medium without the nanomaterials as the control group. And then, the cells were gently rinsed twice for further observation by laser scanning confocal microscope (Nikon inverted research microscope ECLIPSE Ti). The dyes of propidium iodide (PI) and Calein-AM were used to stain the dead and live cells, respectively. The cancer cells were seeded in 96-well plates (8000 HepG2-Red-FLuc and A549 cells/per well) and incubated for 24 h, followed by the treatments of medium alone, CFPB or CFPB@Lipo nanoframes (28 ppm in cobalt ion concentration) for another 24 h. After treatments, the cells were gently rinsed twice and further stained by PI and Calein-AM following standard process. The distribution of dead and live cells was observed by laser scanning confocal microscope.

### The orthotopic hepatocellular carcinoma animal model
The in vivo studies were performed in NOD/SCID nude mice. Nude mice (male, 6–8 weeks old) were purchased from the Laboratory Animal Center of the National Science Council. Animal care was provided in

accordance with the Laboratory Animal Welfare Act and the Guide for the Care and Use of Laboratory Animals and approved by the Institutional Animal Care and Use Committee of Kaohsiung Chang Gung Memorial Hospital (KCGMH). All animal treatments and surgical procedures were performed in accordance with the guidelines of KCGMH Laboratory Animal Center (IACUC NO. 2021031802). The experimental mice were housed in cages (three to five mice in each cage) at 22 ~ 23 °C and 55 ± 10% humidity with 13 h/11 h light/dark cycle. The tumor size is <5 mm$^3$ and at the same time, it will not affect animal physiology. $2 \times 10^6$ Hep G2-Red-Fluc cells resuspended in 20 μL phosphate-buffered saline were surgically implanted into either right or left lobe of the liver with a midline abdominal incision. The mice underwent bioluminescence imaging after implantation. The bioluminescence flux was recorded to assess tumor growth with D-luciferin injection (catalog no. L-8220, Biosynth Carbosynth) and subjected to in vivo imaging system (IVIS, PerkinElmer, Waltham, MA) in bioluminescence mode. Data analysis was performed using Living Image software (version 4.7.3 (PerkinElmer, USA)). The body weight of experimental mice were measured by electronic weighing machine (JENG HENG, Taiwan).

### In vivo orthotopic model in therapeutic studies

The tumor-bearing nude mice were injected via tail vein with 100 ppm per mice of CFPB@Lipo nanocubes or CFPB@Lipo nanoframes while the control group was administered with PBS. Live mice images were acquired with D-luciferin and were subjected to PerkinElmer IVIS Spectrum In Vivo Imaging System in bioluminescence mode at different time points. Bioluminescence flux was recorded to assess tumor growth. The tumor growth inhibition (TGI) rate was calculated with TGI = [1-(treatment group relative tumor volume)/(control group relative tumor volume)] * 100%. Data analysis was performed using Living Image software (version 4.7.3 (PerkinElmer, USA)). A pseudo color image representing the spatial distribution of photon counts were projected onto the photographic image.

### The tumor tissues in DNA damage detection and hypoxia staining

For the morphological analysis, tumor tissues were paraffin-embedded and then sliced into 5 μm thicknesses. The sections were deparaffinized, rehydrated and washed in PBS. The tumor tissues were incubated with phospho-histone H2A.X polyclonal antibody (Invitrogen), then stained by Super Sensitive Polymer-HRP IHC Detection System (BioGenex) according to the manufacturer protocol. For tissue hypoxia staining, the mice were i.p. injected with 1.8 mg of pimonidazole HCl in saline (Hypoxyprobe™-1) 1 h prior to sacrificing. Tumor tissue were paraffin-embedded and then sliced into 4 μm thicknesses. The sections were deparaffinized, rehydrated, washed in PBS, and stained with IgG1 mouse monoclonal antibody. In addition, tumor tissues incubated with phospho-histone H2A.X polyclonal antibody (Invitrogen), then stained by Super Sensitive Polymer-HRP IHC Detection System (BioGenex) according to the manufacturer protocol. The sections were observed by Pannoramic MIDI (3D HISTECH Ltd.).

### In vivo superficial xenograft tumor model and photothermal treatment

The A549 lung carcinoma tumor-bearing superficial xenograft mice were injected intratumorly with 100 ppm per mice of CFPB@Lipo nanoframes while the control group was administered with PBS. After post-injection, 808 nm laser was irradiated with power density 0.8 W/cm$^2$ for 10 min. After laser treatment, the tumor growth was monitored every 2 days until 3 weeks.

### Biodistribution analysis of CFPB@Lipo nanoframes in healthy mice

Male BALB/c mice aged 4–7 weeks were obtained from the Laboratory Animal Center at National Cheng Kung University, Taiwan. The CFPB@Lipo nanoframes, with a Co concentration of 100 ppm per mice, were injected via the tail vein into the healthy BALB/c mice ($n = 3$ per group), with PBS being used as the control group. The main organs of the mice (heart, liver, spleen, lungs, and kidneys) were then harvested and analyzed by atomic absorption spectroscopy (AA) to determine Co content.

### Hematoxylin and eosin (H&E) staining

The organ samples (heart, lung, spleen, liver, and kidney) were embedded in paraffin and cut into slices with a thickness of 5 μm. These sections were then deparaffinized, rehydrated, and washed in PBS. Subsequently, they were stained with hematoxylin solution for 3 min and rinsed in tap water. Following this, the sections were stained with eosin solution for 1 min. To complete the process, the sections were immersed in ethanol, followed by xylene, and then mounted for evaluation. Microscopic observation of the sections was performed using an Olympus BX51 microscope (Olympus, Japan), and three different fields were captured for each group.

### First-principles DFT calculations

Plane-wave-based DFT calculations were conducted via the Vienna Ab initio Simulation Package (VASP)[51] and revised Generalized Gradient Approximation (GGA) of Perdew, Burke Ernzerhof for solids (PBEsol)[52], which was shown capable of accurately calculating the materials properties of Prussian blues[53]. The projector augmented wave was used to treat the core–valence electron interaction[54]. The following valence electrons orbitals are selected: $3s^23p^64s^1$ for K, $2S^22P^4$ for O, $3P^63d^74s^1$ for Fe, $3d^84s^1$ for Co, $2s^22p^2$ for C, $2s^22p^3$ for N. Hubbard-type corrections on GGA (GGA + U) were added for Co and Fe to adjust the on-site Coulomb interaction of localized d-orbital electrons. U parameters of 5.5 eV and 3 eV were used for Fe and Co, respectively, based on the previous computational studies[43,53], which agree well in lattice parameters between experiments and computations. Cutoff energy was set at 500 eV, and the Monkhorst–Pack scheme[55] with a k-spacing of $2\pi \times 0.03$ Å$^{-1}$ ($3 \times 3 \times 3$ for bulk and $3 \times 3 \times 1$ for surface) and $2\pi \times 0.01$ Å$^{-1}$ ($10 \times 10 \times 10$ for bulk and $10 \times 10 \times 1$ for surface) were applied for geometry optimization and PDOS calculations, respectively. Gaussian smearing with a width of 0.1 eV was used for the geometry optimization, while the tetrahedron method with Blöchl corrections was used for PDOS calculations to improve accuracy. Spin-polarization was considered through calculations. The convergence criteria for the electronic minimization and ionic step are set at $10^{-5}$ eV in energy and 0.03 eV/Å in force. Both the lattice and atom positions were relaxed to release the stress. While for surface geometry optimizations, only the atoms are allowed to move. The detailed constructions of atomic bulk and surface structures can be found in Supplementary Information.

The work function of different CFPB follows the procedures proposed before[56] :The average static potential and Fermi level in the bulk structures are calculated first, as shown in Supplementary Fig. 24. Then, the electrostatic potential and the vacuum level of the surface structures are calculated and shown in Supplementary Fig. 25. The work function is the difference between the vacuum level and the Fermi level in the surface structures. We add the difference between the potential and the Fermi level in the bulk structure to the average potential in the surface structure to obtain the Fermi level, and then the work function can be obtained.

### Reporting summary

Further information on research design is available in the Nature Portfolio Reporting Summary linked to this article.

## Data availability

All data generated that support the findings of this study are present in the article and supplementary information. The full image dataset is

available from the corresponding author upon request. Besides, source data are provided with this paper.

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

## Acknowledgements

C.S. Yeh appreciates the financial support by the Ministry of Science and Technology, Taiwan (MOST109-2113-M-006-011-MY3). This research was also supported in part by Higher Education Sprout Project, Ministry of Education to the Headquarters of University Advancement at National Cheng Kung University. Additional financially support was provided by the Center of Applied Nanomedicine, National Cheng Kung University under the Featured Areas Research Center Program within the framework of the Higher Education Sprout Project of the Ministry of Education (MOE) in Taiwan. W. P. Li appreciates the financial support by the National Science and Technology Council, Taiwan (NSTC 109-2113-M-037-017-MY3, NSTC 112-2320-B-037-035 and NSTC 112-2113-M-037-014-MY2), the Kaohsiung Medical University Research Foundation (KMU-Q111002) and the Yushan Young Scholar Program under the MOE, Taiwan. C. H. Su appreciates the financial support by the National Science and Technology Council, Taiwan (NSTC 111-2811-B-182-033-, 109-2314-B-182-082-MY3, and 112-2314-B-182-045-MY3), and the Chang Gung Medical Foundation, Taiwan (CMRPG8M0121-3 and CMRPD5N0011-3). H.K.T appreciates the financial support from the National Science and Technology Council of Taiwan (110-2222-E-006-014-MY3) and the National Center for High-performance Computing (NCHC) for providing computational and storage resources.

## Author contributions

C.S.Y. and W.P.L. conceived the research, interpreted data, and wrote manuscript. C.S.Y. and W.P.L. designed experiments. H.K.T. performed simulation studies and wrote manuscript. C.H.S. designed the experiments, interpreted data and wrote manuscript. L.C.W. designed experiments, materials preparation and characterization and cells study. P.Y.C. and Y.P.H. prepared materials and characterization, cells imaging and animal studies, C.H.H. Y.H.W. and G.L.H. prepared materials and characterization. Y.C.C. conducted flow cytometry. L.C.C. and W.P.S. performed tissues staining. D.M. helps for manuscript editing. W.J.W. performed in vitro experiments. C.L.L., M.C.L. and S.T. performed in vitro, in vivo, and ex vivo optical imaging.

## Competing interests

The authors declare no competing interests.
