## [Peer Review File · Nature Communications]

Reviewers' Comments:

Reviewer #1:

Remarks to the Author:

This work reported stable water oxidation nanoparticles, allowing for the sustainable, external energy-free self-supply of $\cdot\text{OH}$ for chemodynamic (CDT) or photothermal (PTT) therapy, suggesting some attractive design of the nanocube and nanoframe. Surprisingly, both the CFPB nanocube and nanoframe allow for self-supply of O_2 , H_2O_2 and $\cdot\text{OH}$ and it indicated some novel phenomenon on the oxidation of water H_2O ($\text{H}_2\text{O} \rightarrow \text{O}_2 \rightarrow \text{H}_2\text{O}_2 \rightarrow \cdot\text{OH}$). However, some concerns still need to be addressed before publication.

1. For the morphological changes, the author needs to give more details, including the element quantification according to the different times in Figure b-g, and the ICP result of the nanocube and nanoframe should be provided.
2. Regarding the PTT property, what are the differences in the absorbance between the CFPB nanocubes and CFPB nanoframes? Please provide it. And the CFPB photothermal conversion efficiency η of CFPB nanoframes under irradiation (808 nm) is 13.6%, which is not higher than the average photothermal conversion efficiency in many previously reported ones. Please provide a reason with more discussion.
3. The compositional figure 2a and 2b of the CFPB in the figure are not clear, and please give the quantification with the error bar of the XRD analysis.
4. As for the oxygen generation, does the author need to explain the oxidation activation? The Figure 3a and 3b at 0 min should be described. More generally, what is the activation mechanism of CFPB for the production of O_2 to H_2O_2 ? The author should add other experiments, including some negative and positive control groups, to explain the process and mechanism.
5. As the author showed, no obvious $\cdot\text{OH}$ was detected within CFPB and CFPB@Liponanoframes, which is a little bit not according to the high efficiency of water oxidation. Please explain how to prevent the generation of the additional $\cdot\text{OH}$ in blood circulation. And the authors need to clarify the content of each component in CFPB@Lipo.
6. As for the in vivo experiment distribution and metabolism, the distribution was detected by the Rhodamine (RB) encapsulated NPs, which may not be very accurate due to the release of the RB. Other results, such as the ICP test with different times, should be provided to confirm the distribution and metabolism of the NP.
7. As for the tumor inhibition experiment, the quantification of O_2 and $\cdot\text{OH}$ within the tumor, liver, and kidney should be provided in detail. On the other hand, the control group in figure 6a-f should be provided, and in vivo blood biocompatibility should be provided.

Other revision:

1. The authors should add some new articles with more discussions as references.
2. The resolution of the picture is not enough, and the authors should provide clearer ones.
3. The authors need to provide significant analysis of each quantification result.

Reviewer #2:

Remarks to the Author:

The authors present an interesting manuscript about the synthesis and characterization of a novel nanoframe based on Prussian blue (PB) for photothermal therapy. In the last few years, PB has generated a great interest among the scientific community devoted to the materials' field and particularly, for its potential application in theranostics. Therefore, the subject of the study is of interest not only in the biomedical field but also in related subjects. In addition, the results are supported with several techniques and the methodology is acceptable. However, despite the quality of the work, there are several aspects that should be addressed before its publication.

1. The title should be more concise. It is not attractive neither clear.
2. Liposome preparation (from lines 556) is not accurately described. Why were DOPC and DOPE chosen? Their ratio should be expressed as molar ratio.
3. The lipid film is not dissolved but dispersed after hydration and shaking.

4. It also remains unclear the role of the lipids. What is the need of preparing liposomes to be added to PB? Why not to simply functionalize PB surface with lipids?
5. There is no data assessing the mixture of lipids and PB in the media. Since lipid-based structures are essential in the study, it would be convenient to improve this section.

Reviewer #3:

Remarks to the Author:

In this manuscript, the authors designed a water oxidation CoFe Prussian blue (CFPB) nanoframe to realize sufficient external $\cdot\text{OH}$ production and ex nihilo NIR absorption to achieve the synergistic antitumor effect of CDT & PTT. This study provided comprehensive data for characterization and mechanism studies. However, the manuscript still needs major revisions before acceptance. Below are some issues that need to be addressed.

Major :

1. Three consecutive sentences in the second paragraph of the introduction are highly repetitive with the abstract, so it is suggested to make adjustments or supplements. (From "Unexpectedly, a CoFe Prussian blue (CFPB) nanocube" to " and Fenton-like reactions from CFPB")
2. At the end of the introduction section, a brief summary of the main work of the article is required.
3. The nanoframe outperformed in the $\text{H}_2\text{O}_2 \rightarrow \cdot\text{OH}$ process, but the preceding steps seemed to be more efficient for nanocube, is there a comparison of the catalytic efficiency of the whole process ($\text{H}_2\text{O} \rightarrow \cdot\text{OH}$)?
4. An orthotopic HCC mouse model was used in the animal experimental section, so why was a A549 lung carcinoma cells used in the cell experiments?
5. Figure 6g shows that complete tumor suppression was achieved with only one treatment, whereas this effect was not achieved in deep tumors (without PTT), yet such a significant difference was not seen in the cellular results (with or without PTT), more evidence is needed to validate the results of animal studies.

Minor:

1. The language of the manuscript should be improved. Some sentences could be refined to eliminate repetitive words, and the linking words between sentences are sometimes used inappropriately.
2. When discussing results, it is better to use a uniform tense.
3. A series of SEM images can be assembled into one with the same serial number (Figure 1b-g).
4. In Figure 1l and 1m, what is the concentration used for the heating performance and UV-Vis spectrum measurement?
5. Y-axis is missing in Figure 2c
6. In Figure 5b,c,d,f, the scale bar need to be noted.
7. More analysis of the figures could be discussed in the result section rather than in the figure annotations.

REVIEWER COMMENTS

Reviewer #1 (Remarks to the Author):

This work reported stable water oxidation nanoparticles, allowing for the sustainable, external energy-free self-supply of $\cdot\text{OH}$ for chemodynamic (CDT) or photothermal (PTT) therapy, suggesting some attractive design of the nanocube and nanoframe. Surprisingly, both the CFPB nanocube and nanoframe allow for self-supply of O_2 , H_2O_2 and $\cdot\text{OH}$ and it indicated some novel phenomenon on the oxidation of water H_2O ($\text{H}_2\text{O} \rightarrow \text{O}_2 \rightarrow \text{H}_2\text{O}_2 \rightarrow \cdot\text{OH}$). However, some concerns still need to be addressed before publication.

1. For the morphological changes, the author needs to give more details, including the element quantification according to the different times in Figure 1 b-g, and the ICP result of the nanocube and nanoframe should be provided.

RESPONSE: We sincerely appreciate the reviewer's comments. The compositional element quantification of nanoparticles at different reaction times has been provided in the original Fig. 1h, shown as Co/Fe ratios. Currently, this quantification analysis has been updated by adopting three independent results with deviation information, as shown in Fig. R1 (Fig. 1h in the revised manuscript). These data were obtained by the atomic absorption spectroscopic (AA) measurements, which are available in our Department and provide the same quantitative values as the ICP. In addition, the measured raw data of Co/Fe elements in CFPB nanoparticles as the function of etching reaction time have also been provided in the new Supplementary Table 1.

Figure R1. The Co/Fe ratios determined by AA measurements as a function of the etching duration for Fig. 1b-g.

2. Regarding the PTT property, what are the differences in the absorbance between the CFPB nanocubes and CFPB nanoframes? Please provide it. And the CFPB photothermal conversion efficiency η of CFPB nanoframes under irradiation (808 nm) is 13.6%, which is not higher than the average photothermal conversion efficiency in many previously reported ones. Please provide a reason with more discussion.

RESPONSE: We appreciate the reviewer's comments. The PB containing $\text{Fe}^{3+}\text{-N}\equiv\text{C-Fe}^{2+}$ composition shows the featured absorbance at the NIR region (600-900 nm), which is attributed to the photo-induced charge transfer from Fe^{2+} to Fe^{3+} . However, no NIR absorption feature was detected from CFPB nanocubes because they do not contain $\text{Fe}^{3+}\text{-N}\equiv\text{C-Fe}^{2+}$ composition. CFPB nanocubes are CoFe Prussian blue nanocubes. During the acid-etching process, the NIR absorbance of colloids gradually increases as a function of reaction time, suggesting the gradual increase of $\text{Fe}^{3+}\text{-N}\equiv\text{C-Fe}^{2+}$ composition in the products yielding CFPB nanoframes. Fig. 1j in the revised manuscript (below Fig. R2) reveals a significant difference in NIR absorbance between CFPB nanocubes and CFPB nanoframes. For a better understanding, we have labeled 0 h: CFPB nanocubes and 24 h: CFPB nanoframes in Fig. 1j (below Fig. R2).

Figure R2. UV-Vis spectrum of the nanostructures as a function of the etching duration. Inset: Photograph depicting the color of colloidal solutions following etching duration.

The photothermal conversion efficiency η of the materials is a comparable index to predict their photothermal effects. Due to the innate feature of each material, the different photothermal materials reveal their own reasonable η range. Causing different η values from the same material might be attributed to the effects of different preparation methods, particle size, dispersity, surface characterization, and the error in η measurements. For example, PB as a well-

known photothermal agent has a general η range of 10-23 % [ACS Nano, 10(12), 11027-11036. (2016); RSC Advances, 4(56), 29729-29734. (2014); Nature communications, 10(1), 4490. (2019)]. In our case, the η value of CFPB nanoframes was calculated as 13.6%, which is in the reasonable range. As requested by reviewer, we have added the description regarding PB's optical feature. Please see p. 3.

3. The compositional figure 2a and 2b of the CFPB in the figure are not clear, and please give the quantification with the error bar of the XRD analysis.

RESPONSE: We sincerely thank the reviewer's suggestions. Because the original images were taken under the energy dispersive X-ray spectroscopy (EDX) model, the resolution is relatively low. Therefore, the revised figures have replaced the old images with high-resolution images to make the clearer features. (below Fig. 3R a,b; Fig. 2a,b in the revised manuscript). For XRD analysis, the newly added quantification with the error bars was provided (below Fig. R4; Fig. 2e in the revised manuscript).

Figure R3. a, Area selected for the elemental analysis of the CFPB after 6 h of reaction. b, Area selected for the elemental analysis of the CFPB after 24 h of corrosion.

Figure R4. The shift of the diffraction angle in 2θ showing the change in the (200), (220), (400), and (420) crystal planes monitored as a function of the etching duration with CFPB nanocubes (0 h) as the starting material.

4. As for the oxygen generation, does the author need to explain the oxidation activation? The **Fig3a and 3b** at 0 min should be described. More generally, what is the activation mechanism of CFPB for the production of O_2 to H_2O_2 ? The author should add other experiments, including some negative and positive control groups, to explain the process and mechanism.

RESPONSE: We sincerely appreciate the reviewer's comments. The water oxidation reaction or oxygen evolution includes four main steps: (1) water association with the catalytic metal site (M), (2) oxidizing water to form $M=O$, (3) reacting with another water molecule for O-O bonding formation, and (4) M releasing oxygen to expose the vacancy for the following reaction cycle [*The Journal of Physical Chemistry C*, 116(10), 6474-6483. (2012)]. In oxygen evolution, the oxidation activation of the catalyst is the most critical factor. According to our simulation results, the $Co^{2+}-N\equiv C-Fe^{3+}$ composition in non-stoichiometric Co deficient and defective CFPB nanocrystals contains unoccupied bands showing higher reduction potential than 1.23 V (Standard Hydrogen Electrode (SHE) potential of water oxidation) that are capable of accepting electrons from water oxidation (Fig. 4 in the revised manuscript). The related description has been

added and can be found in the Discussion section (p.13) in the revised manuscript.

The descriptions of 0 min in Figure 3a-c, 3g have been added. The meaning of 0 min represents the indicator only without NPs. Please see p. 6 in the revised manuscript.

Like the water oxidation reaction, the oxygen reduction reaction (ORR; $O_2 \rightarrow H_2O_2$) also relies on a catalyst to conduct. The process includes the oxygen absorption with the metal catalyst, involving H^+ and an electron transfer from M to form M-O-O-H, and then releasing H_2O_2 by another couple of H^+ /electron [Chem 7, 38-63 (2021)]. Based on the simulation, the Co^{2+} - $N \equiv C$ - Fe^{2+} composition in non-stoichiometric Co rich CFPB nanocrystals shows the band located at a lower reduction potential than 0.7 V (SHE potential of ORR), thus enables the two electrons donation for two O-H bonds formation (Fig. 4). The related description has been added and can be found in the Discussion section (p.13) in the revised manuscript.

We sincerely appreciate the reviewer's suggestions. PB nanoparticles are used as a negative control that show no apparent change of oxygen, hydrogen peroxide, and hydroxyl free radical generation (below Fig. R5; Fig. 3 in the revised manuscript). The results of PB without O_2 , H_2O_2 , and $\bullet OH$ generation are consistent with previous reports [Adv. Mater 34, 2200389 (2022); Chem. Commun. 55, 7151-7154 (2019)].

Finally, we have additionally performed using anhydrous DMSO to replace water for evaluation the role of water in this catalytic reaction (below Fig. R6). No O_2 and $\bullet OH$ were generated from CFPB nanoframes without water. This examination excludes the evaluation of H_2O_2 generation because the KI (H_2O_2 indicator) is not soluble in DMSO.

Figure R5. a, Oxygen generation by water oxidation reaction activated by PB, CFPB nanocubes and CFPB nanoframes. b, H_2O_2 generation by oxygen reduction reaction triggered by PB, CFPB nanocubes and CFPB nanoframes. c, The generation of $\bullet OH$ via Fenton-like reaction initiated by PB, CFPB nanocubes and CFPB nanoframes. The concentrations for these experiments were fixed as 14 ppm of Fe for PB and Co for CFPB.

Figure R6. a, Oxygen generation through water oxidation activated by CFPB nanoframes under water and DMSO conditions. b, The $\bullet OH$ generation by CFPB nanoframes under PBS and DMSO conditions. The concentration of CFPB is 14 ppm of cobalt ions. The BK represents the blank without CFPB.

5. As the author showed, no obvious $\cdot\text{OH}$ was detected within CFPB and CFPB@Lipo nanoframes, which is a little bit not according to the high efficiency of water oxidation. Please explain how to prevent the generation of the additional $\cdot\text{OH}$ in blood circulation. And the authors need to clarify the content of each component in CFPB@Lipo.

RESPONSE: We sincerely thank the reviewer's comments. Our experimental results (Fig. 3h in the revised manuscript) show that CFPB nanoframes have a greater $\cdot\text{OH}$ generation ability than CFPB nanocubes and CFPB@Lipo nanoframes. Compared to CFPB nanocubes, the nanoframes showing superiority in $\cdot\text{OH}$ generation is due to the relatively higher surface area with more active sites exposed in the frame structure than the solid cube. When the nanoframes are covered by liposomes giving CFPB@Lipo, CFPB@Lipo nanoframes reveal significant inhibition in $\cdot\text{OH}$ generation, which is attributed to the phospholipid bilayer membrane on the surface of CFPB nanoframes that can effectively protect CFPB nanoframes, thus terminating the water-driven chemodynamic reaction. According to the previous report, only proton ions are able to cross through the phospholipid membrane freely [*Biochimica et Biophysica Acta (BBA)-Bioenergetics* 1860, 439-451 (2019)]. In addition, the lipid membrane-made nanovesicles reveal excellent stability during blood circulation [*Adv. Healthc. Mater.* 11, 2100639 (2022)]. Therefore, the encapsulation of CFPB nanoframes by liposomes is an appropriate construction to prevent the $\cdot\text{OH}$ generation during blood circulation. Supplementary Fig. 14 in the revised manuscript shows no spontaneous $\cdot\text{OH}$ generation, hemolysis, and cytotoxicity in endothelial cells when treated with CFPB@Lipo nanoframes. The description of phospholipid bilayer membrane has been added in the revised manuscript p. 7.

FTIR spectra of liposomes and CFPB@Lipo nanoframes have been provided in Supplementary Fig. 8 in the revised manuscript, showing a highly overlapping vibration feature of liposome on the CFPB@Lipo nanoframes. In addition, the more detailed characterization before and after of functionalization of CFPB nanoframes were evaluated by zeta potential, DLS, TEM and TGA analysis (below Fig. R7; Supplementary Fig. 8 in the revised manuscript). The zeta potential of CFPB nanoframes is ~ -32 mV. The surface charge of CFPB@Lipo nanoframes becomes more positive (~ -8 mV) after liposome encapsulation, showing a similar zeta potential with liposome (below Fig. R7a). The hydrodynamic diameter of CFPB@Lipo nanoframes apparently increases after modification given the liposome coverage on the surface of CFPB nanoframes (below Fig. R7b). TEM image reveals CFPB@Lipo nanoframes (below Fig. R7c). TGA analysis showing the remaining weight of 37.8 % and 25.5 % for CFPB nanoframes and CFPB@Lipo nanoframes, respectively. From the loss of weight, the quantification

of Lipo was determined to be 12.3%. (below Fig. R7d). These results all indicate the successful coverage of liposomes onto the CFPB nanoframes. The relevant description has been added in the revised manuscript p. 8.

Figure R7. a, The zeta potential analysis of liposome (Lipo), CFPB, and CFPB@Lipo nanoframes. b, The DLS analysis of Lipo, CFPB, and CFPB@Lipo nanoframes. c, TEM image of CFPB@Lipo nanoframes (scale bars is 100 nm). d, TGA analysis showing the weight loss of CFPB@Lipo nanoframes and CFPB nanoframes.

6. As for the in vivo experiment distribution and metabolism, the distribution was detected by the Rhodamine (RB) encapsulated NPs, which may not be very accurate due to the release of the RB. Other results, such as the ICP test with different times, should be provided to confirm the distribution and metabolism of the NP.

RESPONSE: We sincerely thank the reviewer's valuable comments. We have conducted additional Rb release experiments under 10% serum for 3 and 24 h (below Fig. R8). No sign of RB lost was seen in RB-loaded CFPB@Lipo nanoframes after 24 h incubation. As requested by reviewer, the additional biodistribution analysis were performed using atomic absorption spectroscopic (AA) measurements, which are available in our Department and provide the same

quantitative values as the ICP. Following the same scenario for *ex vivo* imaging from the dissected tissues at the post-injection 3 h and 24 h, as shown in Supplementary Fig. 15 in the revised manuscript, the major organs were harvested for CFPB@Lipo nanocubes and CFPB@Lipo nanoframes. The results are consistent with *ex vivo* imaging showing the highest NPs accumulation in the liver and kidneys at 3 h post-injection and significant elimination after 24 h (below Fig. R9). The biodistribution results by IVIS *ex vivo* and AA were all provided in the Supplementary Fig. 15. Please see p.11 in the revised manuscript for The description in biodistribution using AA measurements.

Figure R8. Relative fluorescence intensity at 570 nm of RB-loaded CFPB@Lipo nanoframes (100 ppm) incubated in 10% serum for 3 and 24 h.

Figure R9. Biodistribution analysis of mice treated with CFPB@Lipo nanocubes

and CFPB@Lipo nanoframes at 3 and 24 h post-injection determined by AA measurements. The mice administrated with PBS were selected as the control group. All measurements were repeated three times.

7. As for the tumor inhibition experiment, the quantification of O_2 and $\cdot OH$ within the tumor, liver, and kidney should be provided in detail. On the other hand, the control group in figure 6a-f should be provided, and *in vivo* blood biocompatibility should be provided.

RESPONSE: a. We express our sincere appreciation to the reviewer for the valuable input. However, the absence of the established standard experimental protocols, particularly for *in vivo* systems, to quantify *in vivo* O_2 and $\cdot OH$ levels has constrained our eagerness to pursue further quantification. It is noted that there is a considerable challenge in measuring the short-lived free radicals ($\cdot OH$), particularly in *in vivo* systems. If *ex vivo* is chosen for the alternative approach, the free radicals ($\cdot OH$) would change rapidly due to their short half-life when tissues are harvested from the animals. Recently, Wu *et al.* have applied the diene electrochromic species (1-Br-Et) as a $\cdot OH$ -responsive chromophore in tumors for fluorescent and photoacoustic imaging [*Nat. Commun.* 12, 6145 (2021)]. However, the detection is still limited in the qualitative approach and the signal is quite weak for differentiation/quantification. It is also noted that the 1-Br-Et probe is synthesized by the authors of *Nat. Commun.* 12, 6145 (2021).

For the detection of oxygenation for *in vivo* studies, quantitative susceptibility mapping (QSM) with magnetic resonance (MR) imaging has been employed [*PLoS ONE* 11(3), e0149602 (2016)]. Nevertheless, this imaging technique still provides qualitative assessments of O_2 for *in vivo* assays. Notably, the QSM equipped with MR imaging was applied only in the brain imaging of stroke animals. In addition, the technique using QSM equipped with MR imaging has beyond our expertise. It has been reported that the variation of O_2 levels was investigated for cells and *ex vivo* tissues using phosphorescent lifetime imaging (PLIM) based on the ultra-fast optical imager (Tpx3Cam facility) [*Biomed. Opt. Express* 11(1), 77 (2020); *Molecules*, 26, 2898 (2021)]. We are sorry to mention that the ultra-fast optical technique is beyond our capabilities. Furthermore, the Tpx3Cam ultra-fast optical facility is not available and is not able to access.

b. The control group has been added in Fig. 6 in the revised manuscript and shown below as Fig. R10.

c. Regarding *in vivo* blood biocompatibility, the red blood cells were incubated with CFPB@Lipo nanoframes at various concentrations to conduct the hemolysis assay (Supplementary Fig. 14b in the revised manuscript). The results reveal no

significant hemolysis indicating that the nanoparticles are favorable for *in vivo* blood biocompatibility. In addition, the biochemical blood test results also show no significant difference in liver and kidney function indexes as compared to control group using PBS (Supplementary Fig. 19 in the revised manuscript)

Figure R10. a, The animal bioluminescence images from Hep G2-Red-FLuc cells

were monitored using the IVIS imaging system (scale bars is 1 cm). b, The tumor growth profiles from the different treated groups. c, The tumor growth profiles without control group from b. d, Illustration of the tumor growth inhibition (TGI) rate. e, The variation of body weight from the different treated groups. f, The DNA damage and hypoxia observation of the tumor region in liver tissue. g, Superficial tumor growth curves with diode laser exposure at 0.8 W/cm² for 10 min. P-values were considered statistically significant (**P* < 0.05, ****P* < 0.001).

Other revision:

1. The authors should add some new articles with more discussions as references.

RESPONSE: We sincerely thank the reviewer's suggestions. Some new references have been added in the revised manuscript to justify the related development. The related update is added in the revised manuscript (p. 3, 4).

2. The resolution of the picture is not enough, and the authors should provide clearer ones.

RESPONSE: We appreciate the reviewer's comments. All figures have been rechecked and improved to increase their resolution. Please see the Figure 1-6 in the revised manuscript.

3. The authors need to provide significant analysis of each quantification result.

RESPONSE: We appreciate the reviewer's comments. We have provided statistical analysis for the critical quantification results.

Reviewer #2 (Remarks to the Author):

The authors present an interesting manuscript about the synthesis and characterization of a novel nanoframe based on Prussian blue (PB) for photothermal therapy. In the last few years, PB has generated a great interest among the scientific community devoted to the materials' field and particularly, for its potential application in theranostics. Therefore, the subject of the study is of interest not only in the biomedical field but also in related subjects. In addition, the results are supported with several techniques and the methodology is acceptable.

However, despite the quality of the work, there are several aspects that should be addressed before its publication.

1. The title should be more concise. It is not attractive neither clear.

RESPONSE: We appreciate the reviewer's comments. We have modified the title as "Prussian blue analog from nanocube to nanoframe with separated active sites to catalyze water driven enhanced catalytic treatments".

2. Liposome preparation (from lines 556) is not accurately described. Why were DOPC and DOPE chosen? Their ratio should be expressed as molar ratio.

RESPONSE: We appreciate the reviewer's comments. Actually, several types of phospholipids could be selected to prepare the liposomes showing different surface characteristics, stability, and fusion ability to the cell membranes. DOPC and DOPE are two common options with the highly similar structure to the natural cell membranes. The DOPC featured zero curvature is easy to form the bilayer structure, leading to stable liposomes. On the contrary, using negative curvature DOPE alone to prepare liposomes is not feasible. Formulating DOPC/DOPE-combined liposomes capable of fusion to natural cell membranes is a mature technique [*Curr. Protoc. Immunol.* 120, 14.44. 11-14.44. 21 (2018)]. The ratio of DOPC and DOPE to prepare liposomes in the present study was fixed as 4:1. The relevant description has been added to "Preparation of liposomes" in the Methods section (p. 18) in the revised manuscript.

3. The lipid film is not dissolved but dispersed after hydration and shaking.

RESPONSE: We appreciate the reviewer's comments. We agree with the reviewer's suggestion that the "dispersed" term is more precise than "dissolved" to describe the suspension of lipid film in the buffer solution. We have corrected the related description. Please see "Preparation of liposomes" in the Methods section (p. 18) in the revised manuscript.

4. It also remains unclear the role of the lipids. What is the need of preparing liposomes to be added to PB? Why not to simply functionalize PB surface with lipids?

RESPONSE: We appreciate the reviewer's comments. Our experimental results (Fig. 3h in the revised manuscript) show that CFPB nanoframes have a greater $\bullet\text{OH}$ generation ability than CFPB nanocubes and CFPB@Lipo nanoframes. Compared to CFPB nanocubes, the nanoframes showing superiority in $\bullet\text{OH}$ generation is due to the relatively higher surface area with more active sites exposed in the frame structures than the solid cubes. When the nanoframes are covered by liposomes giving CFPB@Lipo, CFPB@Lipo nanoframes reveal significant inhibition in $\bullet\text{OH}$ generation, which is attributed to the phospholipid bilayer membrane on the surface of CFPB nanoframes that can effectively protect

CFPB nanoframes, thus terminating the water-driven chemodynamic reaction. According to the previous report, only proton ions are able to cross through the phospholipid membrane freely [*Biochimica et Biophysica Acta (BBA)-Bioenergetics* 1860, 439-451 (2019)]. The temporary inhibition of water-driven chemodynamic therapy before the CFPB@Lipo nanoframes reaching the tumor site is necessary to ensure biosafety. The lipid membrane-made nanovesicles reveal excellent stability during blood circulation [*Adv. Healthc. Mater.* 11, 2100639 (2022)]. Therefore, the encapsulation of CFPB nanoframes by liposome is an appropriate construction to prevent the $\bullet\text{OH}$ generation during blood circulation. Supplementary Fig. 14 in the revised manuscript shows no spontaneous $\bullet\text{OH}$ generation, hemolysis, and cytotoxicity in endothelial cells when treated with CFPB@Lipo nanoframes. When the CFPB@Lipo nanoframes reach the tumor, the temporarily silent water-driven reaction of CFPB@Lipo nanoframes could be turned on after entering the cancer cells where the liposome membranes on nanoframes fuse into the cell membranes to expose the surface of nanoframes.

Phospholipid consists of hydrophobic fatty acid chains, glycerol linkage and hydrophilic phosphate head, which suffers from the serious precipitation under water and alcohol systems. Therefore, dichloromethane (CH_2Cl_2) is usually used to dissolve lipid molecules. Unfortunately, CFPB nanoparticles are CH_2Cl_2 -insoluble, resulting in rapid particles aggregation under this hydrophobic solvent. On the other hand, the CFPB nanoparticles coated with liposomes show good stability in PBS medium.

5. There is no data assessing the mixture of lipids and PB in the media. Since lipid-based structures are essential in the study, it would be convenient to improve this section.

RESPONSE: We appreciate the reviewer's comments. The detailed characterization before and after functionalization of CFPB nanoframes with liposomes were evaluated by zeta potential, DLS, TEM and TGA analysis (above Fig. R7; Supplementary Fig. 8 in the revised manuscript). The zeta potential of CFPB nanoframes is ~ -32 mV. The surface charge of CFPB@Lipo nanoframes becomes more positive (~ -8 mV) after liposome encapsulation, showing a similar zeta potential with liposome (above Fig. R7a). The hydrodynamic diameter of CFPB@Lipo nanoframes apparently increase after modification given the liposome coverage on the surface of CFPB nanoframes (above Fig. R7b). TEM image reveals CFPB@Lipo nanoframes (above Fig. R7c). TGA analysis showing the remaining weight of 37.8 % and 25.5 % for CFPB nanoframes and CFPB@Lipo nanoframes, respectively. From the loss of weight, the quantification

of Lipo was determined to be 12.3%. (above Fig. R7d). These results all indicate the successful coverage of liposomes onto the CFPB nanoframes. The relevant description has been added in the revised manuscript p. 7.

Reviewer #3 (Remarks to the Author):

In this manuscript, the authors designed a water oxidation CoFe Prussian blue (CFPB) nanoframe to realize sufficient external $\cdot\text{OH}$ production and ex nihilo NIR absorption to achieve the synergistic antitumor effect of CDT & PTT. This study provided comprehensive data for characterization and mechanism studies. However, the manuscript still needs major revisions before acceptance. Below are some issues that need to be addressed.

Major :

1.Three consecutive sentences in the second paragraph of the introduction are highly repetitive with the abstract, so it is suggested to make adjustments or supplements. (From “Unexpectedly, a CoFe Prussian blue (CFPB) nanocube” to “ and Fenton-like reactions from CFPB”)

RESPONSE: We appreciate the reviewer’s comments. We have rewritten the sentences in the Introduction. Please see p.4 in the revised manuscript.

2.At the end of the introduction section, a brief summary of the main work of the article is required.

RESPONSE: We appreciate the reviewer’s comments. We have included a summary of the main work at the end of the Introduction. Please see p.4 in the revised manuscript.

3.The nanoframe outperformed in the $\text{H}_2\text{O}_2 \rightarrow \cdot\text{OH}$ process, but the preceding steps seemed to be more efficient for nanocube, is there a comparison of the catalytic efficiency of the whole process ($\text{H}_2\text{O} \rightarrow \cdot\text{OH}$)?

RESPONSE: We appreciate the reviewer’s comments. According to the simulation results, the $\text{Co}^{2+}\text{-N}\equiv\text{C-Fe}^{3+}$ composition contributes to the water oxidation reaction for O_2 generation ($\text{H}_2\text{O} \rightarrow \text{O}_2$). As shown in FTIR measurements of Fig. 2c in the revised manuscript, nanocubes exhibit a much more $\text{Co}^{2+}\text{-N}\equiv\text{C-Fe}^{3+}$ composition, which results in a better O_2 generation (Fig. 3b in the revised manuscript). In the oxygen reduction reaction ($\text{O}_2 \rightarrow \text{H}_2\text{O}_2$), the larger amount of O_2 can facilitate H_2O_2

generation in nanocubes. However, the frame structure of nanoframe with greater surface area has promoted H_2O_2 production where nanoframe and nanocube have comparable H_2O_2 amounts (Fig. 3c, d in the revised manuscript). At last step in $\text{H}_2\text{O}_2 \rightarrow \cdot\text{OH}$, nanoframes outperform and have generated the significant amount of $\cdot\text{OH}$ (Fig. 3g in the revised manuscript) due to the composition (more Co^{2+} - $\text{N}\equiv\text{C}$ - Fe^{2+}) and structural (more reactive sites) effects. Overall, the $\text{H}_2\text{O} \rightarrow \text{O}_2 \rightarrow \text{H}_2\text{O}_2 \rightarrow \cdot\text{OH}$ processes performed by nanocubes and nanoframes can be described as follows: (1) For $\text{H}_2\text{O} \rightarrow \text{O}_2$, nanocubes have superior performance in O_2 generation. (2) For $\text{O}_2 \rightarrow \text{H}_2\text{O}_2$, nanoframes produce comparable H_2O_2 to that in nanocubes when reaction begins with less O_2 . (3) For $\text{H}_2\text{O}_2 \rightarrow \cdot\text{OH}$, nanoframes outperform in $\cdot\text{OH}$ production. Taking these results together, nanoframes exhibit greater catalytic efficacy for the the overall $\text{H}_2\text{O} \rightarrow \cdot\text{OH}$ reaction.

4. An orthotopic HCC mouse model was used in the animal experimental section, so why was a A549 lung carcinoma cells used in the cell experiments?

RESPONSE: We appreciate the reviewer's comments. In addition to the results using A549 cell line, we have added the cell experiments using HepG2 cell line (human hepatocellular carcinoma cell line). The results have been provided in Fig. 5 in the revised manuscript and below Fig. R12)

Figure R12. a, Fluorescence intensity of APF representing the level of $\bullet OH$ generated from the control group, CFPB nanoframes, CFPB@Lipo nanoframes and CFPB@Lipo nanocubes under various concentrations over 10 min, 4 h, and 20 h. b, HepG2-Red-FLuc cells treated with CFPB nanoframes, CFPB@Lipo nanoframes, and CFPB@Lipo nanocubes under a 30-min incubation to monitor O_2 generation indicated by $[Ru(dpp)_3]Cl_2$. c, HepG2-Red-FLuc cells treated with CFPB nanoframes, CFPB@Lipo nanoframes, and CFPB@Lipo nanocubes under a 30-min incubation to monitor H_2O_2 generation indicated by DCFH-DA dye (green emissions). d, HepG2-Red-FLuc cells treated with CFPB nanoframes, CFPB@Lipo nanoframes, and CFPB@Lipo nanocubes under a 30-min incubation to monitor $\bullet OH$ generation indicated by APF dye (green emissions). e, Flow cytometry analysis of HepG2-Red-FLuc cancer cells with and without CFPB nanoframes, CFPB@Lipo nanoframes, and CFPB@Lipo nanocubes. f, Live and dead staining for HepG2-Red-FLuc cancer cells treated with CFPB nanoframes, CFPB@Lipo nanoframes, and CFPB@Lipo nanocubes. Significant damage in cells is seen in CFPB@Lipo nanoframes. All data were obtained in triplicate. All scale bars are 50 μm

5. Figure 6g shows that complete tumor suppression was achieved with only one treatment, whereas this effect was not achieved in deep tumors (without PTT), yet such a significant difference was not seen in the cellular results (with or without PTT), more evidence is needed to validate the results of animal studies.

RESPONSE: We appreciate the comments from the reviewer. The particle concentrations should be cautioned in treatments between superficial tumor (PTT) and deep seated tumor (no PTT, but CDT). PTT refers to photothermal therapy and CDT refers to chemodynamic therapy. For superficial tumor, all CFPB@Lipo nanoframes are directly injected into tumors to receive laser irradiation leading to superior tumor inhibition. On the other hand, the orthotopic deep seated tumors are administrated through tail vein injection where the relatively less amount of nanoframes can reach tumor sites through blood circulation. This effect in particle concentration is supported by observing effective tumor suppression in deep seated tumors when the mice receive 2nd dose injection (Fig. 6b,c in the revised manuscript) while the single dose could not suppress efficiently and tumors grow over the course of the day (Supplementary Fig. 16 in the revised manuscript). Please see p.11 in the revised manuscript for the relevant description.

Minor:

1. The language of the manuscript should be improved. Some sentences could be refined to eliminate repetitive words, and the linking words between sentences are sometimes used inappropriately.

RESPONSE: We appreciate the reviewer's comments. The revised manuscript has been proofread by a native English editor.

2. When discussing results, it is better to use a uniform tense.

RESPONSE: We appreciate the reviewer's comments. We have used a uniform tense in the Discussion.

3. A series of SEM images can be assembled into one with the same serial number (Figure 1b-g).

RESPONSE: We appreciate the reviewer's comments. We have performed additional experiments to display SEM for CFPB nanocrystals treated with acid-etching at different times. SEM images have also been assembled with TEM images with the same serial number. Please see Fig. 1 in the revised manuscript and the below Fig. R13.

Figure R13. TEM and SEM images showing the structural transformation of CFPB. All scale bars are 100 nm.

4. In Figure 1l and 1m, what is the concentration used for the heating performance and UV-Vis spectrum measurement?

RESPONSE: We appreciate the reviewer's comments. Fig. 1l and 1m in the original manuscript has been changed to as Fig. 1j and 1k in the revised manuscript. UV-Vis spectroscopy was used to monitor the absorbance change of CFPB nanocrystals during the proton-induced metal replacement reaction. The CFPB nanocubes (0 h) containing 70 ppm of Co were treated with an acid-etching reaction for 0.5, 3, 6, 16, and 24 h subjected to UV-Vis measurements.

Regarding the heating performance in Fig. 1k in the revised manuscript, the concentration of 14 ppm of Co was fixed for CFPB nanocrystals. The 14 ppm of Co concentration has been added in the revised manuscript. Please see p.5 in the revised manuscript.

5. Y-axis is missing in Figure 2c

RESPONSE: We appreciate the reviewer's correction. The "Transmittance (%)" has been added to the y-axis in Fig. 2c in the revised manuscript.

6. In Figure 5b,c,d,f, the scale bar need to be noted.

RESPONSE: We sincerely thank the reviewer's suggestions. All scale bars are 50 μm and have been included in Fig. 5 in the revised manuscript.

7. More analysis of the figures could be discussed in the result section rather than in the figure annotations.

RESPONSE: We sincerely thank the reviewer's suggestions. We had moved some analysis of figures (Fig. 1, Fig.2, Fig.3, and Fig.5 from annotations to the text in the revised manuscript. Please see p.4, 5, 6, 7, 10.

Reviewers' Comments:

Reviewer #1:

Remarks to the Author:

All of my concerns are addressed, and I think the current version is almost available for publication. Some figures are still not clear, such as in figures 2e, 5a, which should be adjusted in advance.

Reviewer #2:

Remarks to the Author:

The authors have successfully addressed the questions risen by the reviewer and can be accepted in the present form.

Reviewer #3:

Remarks to the Author:

The authors have made revisions accordingly. With these improvements and clarification I can now recommend for publication.

Reviewer #1

All of my concerns are addressed, and I think the current version is almost available for publication. Some figures are still not clear, such as in figures 2e, 5a, which should be adjusted in advance.

Response: We thank the reviewer for the positive comments. Figures 2e, 5a have been rechecked and improved to increase their resolution. Please see the Figure 2, 5 in the revised manuscript.

Reviewer #2

The authors have successfully addressed the questions risen by the reviewer and can be accepted in the present form.

Response: We thank the reviewer for the positive comments to the manuscript.

Reviewer #3

The authors have made revisions accordingly. With these improvements and clarification, I can now recommend for publication.

Response: We thank the reviewer for the positive comments to the manuscript for publication in Nature Communications.